# Probabilistic Graph Circuits: Deep Generative Models for Tractable Probabilistic Inference over Graphs

**Milan Papež**[1]        **Martin Rektoris**[1]        **Václav Šmídl**[1]        **Tomáš Pevný**[1]

[1]Artificial Intelligence Center, Czech Technical University, Prague, Czech Republic

## Abstract

Deep generative models (DGMs) have recently demonstrated remarkable success in capturing complex probability distributions over graphs. Although their excellent performance is attributed to powerful and scalable deep neural networks, it is, at the same time, exactly the presence of these highly non-linear transformations that makes DGMs intractable. Indeed, despite representing probability distributions, intractable DGMs deny probabilistic foundations by their inability to answer even the most basic inference queries without approximations or design choices specific to a very narrow range of queries. To address this limitation, we propose probabilistic graph circuits (PGCs), a framework of tractable DGMs that provide exact and efficient probabilistic inference over (arbitrary parts of) graphs. Nonetheless, achieving both exactness and efficiency is challenging in the permutation-invariant setting of graphs. We design PGCs that are inherently invariant and satisfy these two requirements, yet at the cost of low expressive power. Therefore, we investigate two alternative strategies to achieve the invariance: the first sacrifices the efficiency, and the second sacrifices the exactness. We demonstrate that ignoring the permutation invariance can have severe consequences in anomaly detection, and that the latter approach is competitive with, and sometimes better than, existing intractable DGMs in the context of molecular graph generation.

## 1 INTRODUCTION

Graphs form a fundamental framework for modeling relations (edges) between real or abstract objects (nodes) in diverse applications, such as discovering proteins [Ingraham et al., 2019], modeling physical systems [Sanchez-Gonzalez et al., 2020], detecting financial crimes [Li et al., 2023], and searching for neural network architectures [Asthana et al., 2024]. Nonetheless, capturing the probabilistic behavior of even moderately sized graphs can be difficult. While traditional approaches struggle with this problem [Erdos et al., 1960, Holland et al., 1983, Albert and Barabási, 2002], deep generative models (DGMs) have recently proven immensely successful in this respect [Zhu et al., 2022, Guo and Zhao, 2022, Liu et al., 2023b, Du et al., 2024].

**Challenges for graph DGMs.** There are several challenges in designing graph DGMs, including the following ones.

C1. Graphs live in large and complex combinatorial spaces. Indeed, estimated numbers of possible graphs in the molecular domain are enormous [Reymond et al., 2012, Polishchuk et al., 2013]. This poses considerable requirements on the expressivity of DGMs.

C2. Graphs are not random only in values of node and edge features but also in the number of these nodes and edges. Therefore, specific architectures accounting for this variable-size character of graphs are required.

C3. Graphs are permutation invariant, i.e., there is a factorial number of possible configurations of a single graph. The key property of graph DGMs should be to recognize all the configurations as the same graph.

C4. Graphs respect domain-specific semantic validity. For example, not all molecular graphs are chemically valid but must adhere to chemical valency constraints.

**Current solutions.** Depending on the mechanism of generating their outputs, graph DGMs can broadly be divided into *autoregressive* and *one-shot* models. To ensure sufficient expressiveness (C1), both these categories rely on graph neural networks (GNNs) [Wu et al., 2020, Zhang et al., 2020]. To accommodate for the varying size (C2), autoregressive models treat graphs as sequences whose elements are recursively processed by recurrent units [Liao et al., 2019], whereas one-shot models use virtual-node padding [Madhawa et al., 2019] or convolutional GNNs [De Cao and Kipf, 2018]. The permutation invariance (C3) can be imposed in various ways

[Murphy et al., 2019]. Two techniques stand up frequently for both the categories of DGMs. The first relies on sorting the graph into its canonical configuration [Chen et al., 2021], and the second utilizes the permutation equivariance of graph neural networks [Niu et al., 2020]. To capture the semantic validity (C4), both the categories can benefit from domain knowledge, e.g., by using rejection sampling [Shi et al., 2020], post-hoc correction [Zang and Wang, 2020], constraints [Liu et al., 2018], or regularization [Ma et al., 2018]. However, it is preferable to satisfy the validity in a completely domain-agnostic way [Zang and Wang, 2020], relying on the model's expressiveness to extract the essence of the problem.

**The limitation of graph DGMs.** Graph DGMs excel at sampling new graphs, as it is their predetermined probabilistic inference task. However, despite constituting models of a probability distribution, other inference tasks—such as marginalization, conditioning, or expectation—remain elusive. In other words, graph DGMs are *intractable* probabilistic models. There are a few exceptions to that, allowing for one or two more inference tasks beyond mere sampling. Graph normalizing flows [Madhawa et al., 2019] and graph autoregressive models [You et al., 2018] allow for exact likelihood evaluation (conditionally on the canonical configuration of a graph). The autoregressive models additionally provide a narrow range of marginal and conditional distributions, only those respecting the fixed sequential structure of the chain rule of probability. Alternatively, it is a common practice to equip DGMs with extra components explicitly tailored to desired inference tasks, such as conditional sampling [Vignac et al., 2023].

**Tractable probabilistic models.** Probabilistic circuits (PCs) constitute a framework [Choi et al., 2020] which unifies many tractable probabilistic models (TPMs) as special cases. PCs are expressive (C1) deep networks [Martens and Medabalimi, 2015, de Colnet and Mengel, 2021, Yin and Zhao, 2024] that can encode probability distributions with hundreds of millions of parameters [Liu et al., 2023a,c, 2024]. The underlying characteristic of PCs is that they provide exact and efficient answers to a wide range of inference queries—such as marginalization, conditioning, and expectation—without approximations or query-specific architecture modifications. However, their adoption to graphs has received limited attention. Existing PCs for graphs address C2 both in (pseudo-)autoregressive [Errica and Niepert, 2024] or one-shot [Nath and Domingos, 2015, Papež et al., 2024b] manner, adopting the chain rule of probability, where some (or all) of its elements rely on strong independence assumption, whereas other models treat graphs as fixed-size objects [Zheng et al., 2018, Ahmed et al., 2022, Loconte et al., 2023]. To the best of our knowledge, there is currently no PC for graphs satisfying C3, with the exceptions [Nath and Domingos, 2015, Papež et al., 2024b] that are only partially permutation invariant [Diaconis and Freedman, 1984]. These approaches are, however, designed only for specific types of graphs. Moreover, PCs are well-suited to inject domain-specific validity constraints (C4) through knowledge compilation [Ahmed et al., 2022, Loconte et al., 2023]. We offer more remarks on the related work in Section 4.

**Contributions.** The contributions of this paper are summarized as follows.

- To address the limitations of the aforementioned DGMs, we propose probabilistic graph circuits (PGCs), a framework for designing one-shot graph TPMs (that can operate also in the autoregressive regime due to their tractability).
- We extend the standard notion of permutation invariance—which can involve intractable distributions—to a more strict form of *tractable* permutation invariance (C3) and specify the conditions for the tractability of PGCs.
- To address the variable-size character of graphs (C2), we utilize the key benefit of PCs—i.e., their tractability—and propose the *marginalization padding*, which is an approach that marginalizes out non-existing nodes (and associated edges) of input graphs.
- We define conditions for making PGCs inherently and tractably permutation invariant (C3) and discuss that achieving such property can lead to reduced expressive power. Therefore, we investigate two principles that make PGCs permutation invariant even with arbitrary permutation-sensitive building blocks: the first ensures invariance through marginalization, and the second through conditioning on a canonical graph ordering.
- We illustrate the importance of the permutation invariance in a synthetic anomaly detection example. Then, in the context of the unconditional generation of molecular graphs, we show that the order-conditioned PGCs (which we implement using several state-of-the-art variants of PCs and different canonical graph orderings) are competitive—and in most metrics even superior—to intractable graph DGMs. To demonstrate that the PGCs can perform tractable inference tasks efficiently, we generate molecular graphs conditionally on known subgraphs.

## 2 PRELIMINARIES

**Notation.** We denote random variables by upper-case letters, $X$, and their realizations by lower-case letters, $x \in \text{dom}(X)$, where $\text{dom}(X)$ is the domain of $X$. Sets of random variables are expressed by bold upper-case letters, $\mathbf{X} := \{X_1, \dots, X_n\}$, and their realizations by bold lower-case letters, $\mathbf{x} := \{x_1, \dots, x_n\}$. The domain of $\mathbf{X}$ is $\text{dom}(\mathbf{X}) := \text{dom}(X_1) \times \cdots \times \text{dom}(X_n)$. To denote sets of positive integers, we use $[n] := \{1, \dots, n\}$ with $n > 0$. If $\text{dom}(X_i)$ is identical for all $i \in [n]$, we write $\text{dom}(\mathbf{X}) := \text{dom}(X)^n$. We often consider that the elements of $\mathbf{X}$ are organized into a matrix or a tensor such that $\text{dom}(\mathbf{X}) := \text{dom}(X)^{n_1 \times n_2}$ or $\text{dom}(\mathbf{X}) := \text{dom}(X)^{n_1 \times n_2 \times n_3}$, respectively. To index $(i, j, k)$-th entry of a tensor, we use subscripts, i.e., $\mathbf{X}_{ijk}$. To

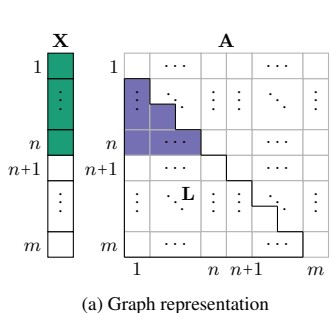

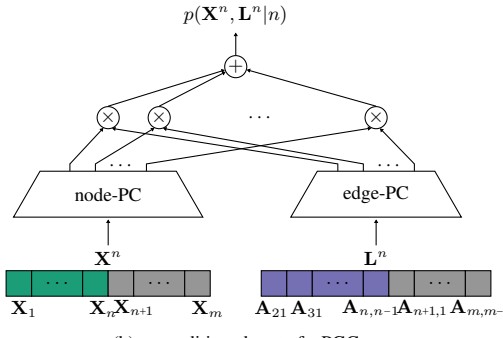

(a) Graph representation

(b) $n$-conditioned part of a PGC

Figure 1: *An example of a PGC for undirected graphs.* (a) We consider a graph $\mathbf{G}$ represented by a feature matrix, $\mathbf{X}$, and an adjacency tensor, $\mathbf{A}$, such that each instance of $\mathbf{G}$ (highlighted in green and blue) has a random number of nodes, $n \in (0, 1, \ldots, m)$, where $m$ is a fixed maximum number of nodes. The empty places (white) are not included in the training data. (b) The main building block of PGCs is the $n$-conditioned joint distribution over $\mathbf{X}^n$ and $\mathbf{L}^n$, the latter of which is a flattened lower triangular part of $\mathbf{A}^n$. $\mathbf{X}^n$ and $\mathbf{L}^n$ are used as input into the node-PC and edge-PC, respectively. The empty places are marginalized out (grey). The outputs of these two PCs are passed through the product layer with $n_c$ units and the sum layer with a single unit.

select slices of a tensor, we use the colon ":", e.g., $\mathbf{X}_{::k}$ is the $k$-th matrix along the third dimension of $\mathbf{X}$. We often use $\mathbf{X}_i \coloneqq \mathbf{X}_{i::}$ or $\mathbf{X}_{ij} \coloneqq \mathbf{X}_{ij:}$ to abbreviate the notation.

Importantly, we consider that all sets of variables are random not only in their values but also in their size, i.e., in $\mathbf{X} \coloneqq \{X_1, \ldots, X_N\}$, not only each $X_i$ is random but $N$ is also random, which we sometimes stress by writing $\mathbf{X} \coloneqq \mathbf{X}^N$. When a set is random only in values but fixed in size, we use $\mathbf{X}^n$. For a realization of values and size, we write $\mathbf{x} \coloneqq \mathbf{x}^n$. The stochastic behavior of a fixed-size random set, $\mathbf{X}^n$, is characterized by a probability distribution, $p(\mathbf{X}^n | n)$, whose structure and the number of parameters depends on $n$. We make this dependence deliberately explicit in this paper. On the contrary, $p(\mathbf{X}) \coloneqq p(\mathbf{X}^N, N)$ is a probability distribution over random values and random size.

**Probabilistic circuits.** PCs are deep computational networks [Vergari et al., 2019a,b] composed of three types of computational units: input, sum, and product units [1]. The key property of PCs is that—under structural constraints [Darwiche and Marquis, 2002]—they are *tractable*, as described in Definition 1.

**Definition 1.** (Tractability of PCs [Choi et al., 2020]). Let $p(\mathbf{X}^n | n)$ be a PC encoding a probability distribution over a fixed-size set, $\mathbf{X}^n$, which is smooth (Assumption 1), decomposable (Assumption 2), and has tractable input units (Assumption 3). Furthermore, consider that $\mathbf{X}^n$ can be organized into two fixed-size subsets, $\mathbf{X}^n \coloneqq \{\mathbf{X}_a^{n-k}, \mathbf{X}_b^k\}$. Then, the integral $\int p(\mathbf{x}_a^{n-k}, \mathbf{X}_b^k | n) d\mathbf{x}_a^{n-k}$ can be computed (**A**) *exactly* (without approximations) and (**B**) *efficiently* with $\mathcal{O}(\text{poly}(|p|))$ complexity, where $|p|$ is the number of connections between the computational units of $p$.

Definition 1 is the key distinguishing feature of PCs. Its main consequence is that many inference tasks [Vergari et al., 2021, Wang et al., 2024] can be performed in a single forward pass through the network.

**Graphs.** We define an $N$-node graph as a tuple, $\mathbf{G} \coloneqq \{\mathbf{X}, \mathbf{A}\}$, containing a node feature matrix, $\mathbf{X}$, and an edge adjacency tensor, $\mathbf{A}$. The domains $\text{dom}(\mathbf{X})$ and $\text{dom}(\mathbf{A})$ are defined by $\text{dom}(X)^{N \times n_X}$ and $\text{dom}(A)^{N \times N \times n_A}$, respectively (where $N$ is random but $n_X$ and $n_A$ are always fixed). To express $\mathbf{G}$ with $n_X$ types of nodes and $n_A$ types of edges, we consider that $\text{dom}(\mathbf{X})$ and $\text{dom}(\mathbf{A})$ are one-hot encoded along the last (fixed-size) dimension. Therefore, it holds that $\text{dom}(X) \coloneqq \{0, 1\}, \sum_{j=1}^{n_X} \mathbf{X}_{ij} = 1$, and $\text{dom}(A) \coloneqq \{0, 1\}, \sum_{k=1}^{n_A} \mathbf{A}_{ijk} = 1$. We assume that $\text{dom}(\mathbf{A})$ includes an extra category to express that there can be no connection between two nodes.

**Graphs are $\mathbb{S}_n$-invariant.** There can be up to $n!$ distinct but equivalent permutations (orderings) of an $n$-node graph, $\mathbf{G}^n$. A properly designed probabilistic model, $p(\mathbf{G}^n | n)$, has to recognize all these configurations by assigning them with the same probability. This property is known as permutation invariance or $\mathbb{S}_n$-invariance. To define $\mathbb{S}_n$-invariance of a probability distribution over a graph, $p(\mathbf{G}^n | n)$, we use a finite symmetric group of a set of $n$ elements, $\mathbb{S}_n$. This is a set of all $n!$ permutations of $[n]$. We consider that each permutation[2], $\boldsymbol{\pi} \in \mathbb{S}_n$, acts upon the first dimension of the feature matrix, $\boldsymbol{\pi} \mathbf{X}^n \coloneqq \{\mathbf{X}_{\pi(1)}, \ldots, \mathbf{X}_{\pi(n)}\}$, and upon the first two dimensions of the adjacency tensor, $\boldsymbol{\pi} \mathbf{A}^n \coloneqq \{\mathbf{A}_{\pi(1)\pi(1)}, \mathbf{A}_{\pi(1)\pi(2)}, \ldots, \mathbf{A}_{\pi(n)\pi(n)}\}$ [Orbanz and Roy, 2014]. This allows us to permute an $n$-node graph, $\mathbf{G}^n \coloneqq (\mathbf{X}^n, \mathbf{A}^n)$, as follows: $\boldsymbol{\pi} \mathbf{G}^n = (\boldsymbol{\pi} \mathbf{X}^n, \boldsymbol{\pi} \mathbf{A}^n)$. We formally introduce the $\mathbb{S}_n$-invariance of $p(\mathbf{G}^n | n)$ in Definition 2.

**Definition 2.** ($\mathbb{S}_n$-invariance). The probability distribution $p(\mathbf{G}^n | n)$ is $\mathbb{S}_n$-invariant iff $p(\boldsymbol{\pi} \mathbf{G}^n | n) = p(\mathbf{G}^n | n)$ for all $\boldsymbol{\pi} \in \mathbb{S}_n$, and, so, $\mathbf{G}$ is $\mathbb{S}_n$-invariant if $p(\mathbf{G}^n | n)$ is.

**Problem definition.** We aim to design a *tractable*, $\mathbb{S}_n$-*invariant* probability distribution over graphs, $p(\mathbf{G})$, and learn its parameters based on a collection of observed graphs,

---

[1] We refer the reader to Appendix B for an introduction to PCs.

[2] Note that we consider a permutation as a set of numbers, $\boldsymbol{\pi} \coloneqq \{\pi(1), \ldots, \pi(n)\}$.

$\{\mathbf{G}_1, \ldots, \mathbf{G}_I\}$, where each $\mathbf{G}_i$ can have a different number of nodes and edges.

# 3 PROBABILISTIC GRAPH CIRCUITS

Conventional PCs encode a tractable probability distribution over a fixed-size set, $p(\mathbf{X}^n|n)$; however, they are not $\mathbb{S}_n$-invariant [Papež et al., 2024b]. In contrast, we propose probabilistic graph circuits that encode a probability distribution over a random-size graph, $p(\mathbf{G})$, considering different ways to ensure their $\mathbb{S}_n$-invariance and its impact on the tractability. We aim that $p(\mathbf{G})$ inherits the ability to answer the broad range of probabilistic inference queries from conventional PCs. To introduce (probabilistic) graph circuits, we start with defining the scope of a graph, which will be important in describing their inner mechanisms.

**Definition 3** (Graph Scope). The scope of $\mathbf{G}$ is an arbitrary subset, $\mathbf{G}_u \subseteq \mathbf{G}$, such that $\mathbf{X}_u \subseteq \mathbf{X}$ and $\mathbf{A}_u \subseteq \mathbf{A}$, i.e., it can be a subgraph, $\mathbf{G}_u = (\mathbf{X}_u, \mathbf{A}_u)$, a set of nodes, $\mathbf{G}_u = \mathbf{X}_u$, or a set of edges, $\mathbf{G}_u = \mathbf{A}_u$.

Note that if we split $\mathbf{G}$ into two scopes $\mathbf{G} := \{\mathbf{G}_a, \mathbf{G}_b\}$, then both $\mathbf{G}_a$ and $\mathbf{G}_b$ have a random number of nodes such that $N = N_a + N_b$.

Graph circuits are syntactically similar to conventional circuits in that they comprise sum, product, and input units. However, the key difference is that these units are defined over variable-size subgraphs (or graph subsets) rather than fixed-size subsets, as introduced in Definitions 4 and 5.

**Definition 4** (Graph Circuit). A *graph circuit* (GC) $c$ over a random graph $\mathbf{G} := (\mathbf{X}, \mathbf{A})$ is a parameterized computational network encoding a function $c(\mathbf{G})$. It contains three types of computational *units*: *input*, *product*, and *sum* units. All sum and product units receive the outputs of other units as inputs. We denote the set of inputs of a unit $u$ as $\text{in}(u)$. Each unit $u$ encodes a function $c_u$ over a subgraph, $\mathbf{G}_u \subseteq \mathbf{G}$, the *graph scope* (Definition 3). The input unit $c_u(\mathbf{G}_u) := f_u(\mathbf{G}_u)$ computes a user-defined, parameterized function, $f_u$; the sum unit computes the weighted sum of its inputs, $c_u(\mathbf{G}_u) := \sum_{i \in \text{in}(u)} w_i c_i(\mathbf{G}_i)$, where $w_i \in \mathbb{R}$ are the weight parameters; and the product unit computes the product of its inputs, $c_u(\mathbf{G}_u) := \prod_{i \in \text{in}(u)} c_i(\mathbf{G}_i)$. The scope of any sum or product unit is the union of its input scopes, $\mathbf{G}_u = \bigcup_{i \in \text{in}(u)} \mathbf{G}_i$.[3] A circuit can have one or multiple root units. The scope of a root unit is $\mathbf{G}$.

**Definition 5** (Probabilistic Graph Circuit). A *probabilistic graph circuit* (PGC) over a random graph $\mathbf{G}$ is a GC (Definition 4) $c$, such that $\forall \mathbf{g} \in \text{dom}(\mathbf{G}) : c(\mathbf{g}) \geq 0$, i.e., it is a *non-negative* function for all realizations of $\mathbf{G}$.

---

[3]For example, making the random number of nodes of $\mathbf{G}$ explicit, the union of scopes for a product unit with two children is $\mathbf{G}^N = \mathbf{G}_a^{N_a} \cup \mathbf{G}_b^{N_b}$, where $N = N_a + N_b$.

A PGC (Definition 5) encodes a possibly unnormalized, joint probability distribution over the size and values of $\mathbf{G}$. Definition 3 implies that the scope of each computational unit of a (P)GC has a random number of nodes (and edges). This random character of computational units opens up for various constructions of PGCs. In this paper, we start to investigate PGCs that take the following form:

$$p(\mathbf{G}) = p(\mathbf{G}^N, N) = p(\mathbf{G}^n|n)p(N), \qquad (1)$$

where $p(\mathbf{G}^n|n)$ is an $n$-conditioned PGC over an $n$-node graph, $\mathbf{G}^n$, which is fixed in its size but describes stochastic behavior of values that the nodes and edges of $\mathbf{G}$ can take; and $p(N)$ is a cardinality distribution, which characterizes the random size of $\mathbf{G}$.

## 3.1 TRACTABILITY OF PGCS

One of the key requirements for GCs is that they should satisfy $\mathbb{S}_n$-invariance (C3). However, the standard definition of $\mathbb{S}_n$-invariance (Definition 2) is too general. Namely, it does not specify how $p$ achieves this property and, especially, whether $p$ is tractable. Tractability of $\mathbb{S}_n$-invariant functions in the deep learning literature takes only the perspective of the computational efficiency [Murphy et al., 2019]. Therefore, we extend Definition 2 with both parts (A and B) of Definition 1 that are canonical in the circuit literature. The PGC (1) is $\mathbb{S}_n$-invariant only if its $n$-conditioned part is $\mathbb{S}_n$-invariant. We thus describe the tractable $\mathbb{S}_n$-invariance in terms of $p(\cdot|n)$, as provided in Definition 6.

**Definition 6.** (Tractable $\mathbb{S}_k$-invariance.) Let $p(\mathbf{G}^n|n)$ be an $n$-conditioned, smooth (Assumption 1) and decomposable (Assumption 2) PGC whose input units admit tractable integration (Assumption 3). Furthermore, consider that $\mathbf{G}^n$ is organized into two *fixed-size* subgraphs, $\mathbf{G}^n = \{\mathbf{G}_a^{n-k}, \mathbf{G}_b^k\}$. Then, $p(\cdot|n)$ is *tractably* $\mathbb{S}_k$-invariant if

$$\int p(\mathbf{g}_a^{n-k}, \boldsymbol{\pi}_b \mathbf{G}_b^k|n)d\mathbf{g}_a^{n-k} = \int p(\mathbf{g}_a^{n-k}, \mathbf{G}_b^k|n)d\mathbf{g}_a^{n-k}$$

can be computed *exactly* in $\mathcal{O}(\text{poly}(|p|))$ time for all $\boldsymbol{\pi}_b \in \mathbb{S}_k$, where $d\mathbf{g}_a^{n-k}$ is a suitable reference measure corresponding to the $(n\text{-}k)$-node realization $\mathbf{g}_a^{n-k}$.

Tractable $\mathbb{S}_k$-invariance results in tractable $\mathbb{S}_n$-invariance for $k = n$. Although there is then no integral to compute in Definition 6, this case implies that the full evidence query $p(\mathbf{g}^n|n)$ is obtained *exactly* in $\mathcal{O}(\text{poly}(|c|))$ time. In other words, we obtain the $p(\mathbf{g}^n|n)$ value in a single forward pass through the circuit. Under this $k = n$ setting, we are backward compatible to Definition 2, yet now we specify what tractability means in this context.

The tractable $\mathbb{S}_k$-invariance (Definition 6) is the key element in the tractability of PGCs. However, the tractability of (1) does not depend only on the $n$-conditioned part, $p(\cdot|n)$, but also on the cardinality distribution, $p(N)$, as specified in Proposition 1.

**Proposition 1.** *(Tractability of PGCs.) Let $p$ be a PGC (1) such that $p(\mathbf{G}^n|n)$ is tractably $\mathbb{S}_n$-invariant (Definition 6), and $p(N)$ has a finite support. Furthermore, consider that $\mathbf{G}$ is organized into two random-size subgraphs, $\mathbf{G} = \{\mathbf{G}_a, \mathbf{G}_b\}$. Then, $p(\mathbf{G})$ is tractable if*

$$\int p(\mathbf{g}_a, \mathbf{G}_b)d\mathbf{g}_a = \sum_{n=k}^{\infty} \int p(\mathbf{g}_a^{n-k}, \mathbf{G}_b^k|n)p(n)d\mathbf{g}_a^{n-k}$$

*can be computed exactly in $\mathcal{O}(poly(|p|))$ time.*

Proposition 1 invokes an impression that the infinite sum makes the integral intractable. However, $p(n)$ is assumed to have finite support, i.e., it is always occupied with non-zero values only in a specific interval, $0 < n \leq m$, while being equal to zero outside. For this reason, the infinite sum becomes a finite one, and the integral is tractable.

**Marginalization padding.** The conventional PCs model probability distributions over fixed-size vectors. However, each instance of $\mathbf{G}$ can have a different number of nodes and edges (C2). The PGC (1) partially reflects the variable-size character of $\mathbf{G}$ by $p(N)$, which describes the random size of the $N$-node graph, $\mathbf{G}^N$. Still, the $n$-node (fixed-size) argument $\mathbf{G}^n$ in $p(\cdot|n)$ raises the question: how to design $p(\cdot|n)$—which should be characterized by a single, fixed-size set of parameters—such that it can take graphs with different values of $n$ as input (e.g., such that we can easily enumerate the sum terms in Proposition 1)? To solve this problem, we utilize the key feature of PGC—their tractability—to propose *marginalization padding*. This mechanism assumes that there can be at most $m$ nodes in a random-size graph, $\mathbf{G}$, and whenever there is an instance with $n < m$ nodes, it marginalizes out the remaining $m - n$ 'empty' nodes and associated edges. We illustrate this principle on a specific instance of PGCs in Figure 1.

## 3.2 INHERENT $\mathbb{S}_n$-INVARIANCE

There are different ways to design a PGC that satisfies Definition 6. The primary question we ask is how to design a PGC that is *inherently* $\mathbb{S}_n$-invariant? The inherent $\mathbb{S}_n$-invariance means that a PGC satisfying Assumptions 1-3 is tractably $\mathbb{S}_n$-invariant without resorting to any external $\mathbb{S}_n$-invariance mechanism. The following definition states the conditions for such a PGC.

**Definition 7.** (Inherently $\mathbb{S}_n$-invariant PGCs.) A tractable PGC (Proposition 1) is inherently $\mathbb{S}_n$-invariant if its computational units satisfy the following conditions:

a) $p_u(\boldsymbol{\pi}_u\mathbf{G}_u) := f_u(\boldsymbol{\pi}_u\mathbf{G}_u)$,
b) $p_u(\boldsymbol{\pi}_u\mathbf{G}_u) = \sum_{i\in\mathsf{in}(u)} w_i p_i(\boldsymbol{\pi}_u\mathbf{G}_i)$,
c) $p_u(\boldsymbol{\pi}_u\mathbf{G}_u) = \prod_{i\in\mathsf{in}(u)} p_i(\boldsymbol{\pi}_i\mathbf{G}_i)$,

for all $\boldsymbol{\pi}_u \subseteq \boldsymbol{\pi} \in \mathbb{S}_n$, where $\boldsymbol{\pi}_i \subseteq \boldsymbol{\pi}_u$ are pairwise disjoint partitions of $\boldsymbol{\pi}_u$. Recall that we treat $\boldsymbol{\pi}$ as a set (Section 2).

The input unit a) is $\mathbb{S}_n$-invariant if $f_u$ is a user-defined $\mathbb{S}_n$-invariant distribution. The sum unit b) is $\mathbb{S}_n$-invariant if all its input units are $\mathbb{S}_n$-invariant. However, it is generally difficult to fulfill the $\mathbb{S}_n$-invariance of the product unit c). Even if the inputs $p_i$ of c) are $\mathbb{S}_n$-invariant, then c) is only *partially* $\mathbb{S}_n$-invariant [Papež et al., 2024b]. The partial invariance means that the subgraph $\mathbf{G}_u$ can be permuted only in its individual partitions $\{\mathbf{G}_i\}_{i\in\mathsf{in}(u)}$. Definition 7(c) is stricter in this sense, as it requires that $\mathbf{G}_u$ can also be permuted among its partitions (i.e., permuting the nodes and the corresponding edges between $\mathbf{G}_v$ and $\mathbf{G}_w$ for $v, w \in \mathsf{in}(u)$).

To find an inherently $\mathbb{S}_n$-invariant PGC, we consider that PCs are discrete latent variable models where each sum unit induces a categorical latent variable over its inputs [Peharz et al., 2016, Zhao et al., 2016, Trapp et al., 2019]. The consequence of Definition 7 is that the latent representation of a PGC must be $\mathbb{S}_n$-invariant for all its components.

**Proposition 2.** *(Latent representation of inherently $\mathbb{S}_n$-invariant PGCs.) A tractable PGC (Proposition 1) in its latent representation,*

$$p(\mathbf{G}) = \sum_{\mathbf{z}\in\mathsf{dom}(\mathbf{Z})} p(\mathbf{G}|\mathbf{z})p(\mathbf{z}), \qquad (2)$$

*is inherently $\mathbb{S}_n$-invariant if $p(\boldsymbol{\pi}\mathbf{G}|\mathbf{z}) = p(\mathbf{G}|\mathbf{z})$ for all $\boldsymbol{\pi} \in \mathbb{S}_n$ and all $\mathbf{z} \in \mathsf{dom}(\mathbf{Z})$, where $\mathbf{Z}$ are discrete latent variables of all sum units.*

One example of an inherently $\mathbb{S}_n$-invariant PGC—which satisfies Proposition 2—can be obtained by assuming that the slices of $\mathbf{X}$ and $\mathbf{A}$ are conditionally independent and identically distributed (i.i.d.), as formulated in Proposition 3.

**Proposition 3.** *(Inherently $\mathbb{S}_n$-invariant PGCs through the conditional i.i.d. assumption.) A tractable PGC (Proposition 1) is inherently $\mathbb{S}_n$-invariant if its components are factorized as*

$$p(\mathbf{G}^n|\mathbf{z}, n) = \prod_{i\in[n]} p(\mathbf{X}_i|\mathbf{z}, n) \prod_{j\in[n]} p(\mathbf{A}_{ij}|\mathbf{z}, n). \qquad (3)$$

Note that the PGC from Proposition 3 captures the correlations both between and within the nodes and edges through the sum in (2), i.e., by conditioning on possibly exponentially many $\mathbf{z} \in \mathsf{dom}(\mathbf{Z})$. However, the i.i.d. assumption means that the product terms in (3) share the parameters for all the slices of $\mathbf{X}$ and, separately, for all the slices of $\mathbf{A}$. Naturally, this sharing mechanism yields less expressive power than having distinct parameters for all the slices. We visualize this PGC in Appendix E (Figure 5). Note also that this shared parameterization must be unique for each component $p(\mathbf{G}^n|\mathbf{z}, n)$, i.e., for each $\mathbf{z} \in \mathsf{dom}(\mathbf{Z})$; otherwise, the expressive power would drop even further by merging duplicate components of (2) into a single one. This strict construction prevents adopting most of the recent design

patterns that share the parameters across the components [Peharz et al., 2020a, Loconte et al., 2024b,a]. Therefore, we further consider strategies to ensure $\mathbb{S}_n$-invariance (Definition 2) even with these traditional $\mathbb{S}_n$-sensitive PCs, though at the cost of not fulfilling Definition 6, as explained next.

## 3.3  $\mathbb{S}_n$-INVARIANCE BY MARGINALIZATION

The general way to achieve $\mathbb{S}_n$-invariance is to augment $p(\mathbf{G}^n|n)$ by a latent random variable, $\boldsymbol{\pi} \in \mathbb{S}_n$, and compute the marginal probability distribution,

$$p(\mathbf{G}^n|n) \coloneqq \sum_{\boldsymbol{\pi} \in \mathbb{S}_n} p(\mathbf{G}^n|\boldsymbol{\pi}, n)p(\boldsymbol{\pi}|n), \qquad (4)$$

which takes $\mathbf{G}^n$ and evaluates the same distribution, $p(\cdot|\boldsymbol{\pi}, n)$, for all orderings $\boldsymbol{\pi} \in \mathbb{S}_n$ of $\mathbf{G}^n$. The marginalization ensures that (4) satisfies the standard definition of $\mathbb{S}_n$-invariance (Definition 2) for arbitrary permutation-sensitive $p(\cdot|\boldsymbol{\pi}, n)$ and uniform $p(\boldsymbol{\pi}|n)$. Now, consider that $p(\cdot|\boldsymbol{\pi}, n)$ follows Assumptions 1-3, and that the summation in (4) is just a sum unit. Then, (4) yields exact values and thus complies with Definition 1(A). However, the factorial complexity of the sum violates the polynomial requirement of Definition 1(B). For this reason, this approach does not adhere to the tractable $\mathbb{S}_n$-invariance (Definition 6).

**Asymmetry implies intractability.** One possible way to tame the factorial complexity of (4) is to simply reduce the number of sum terms [Wagstaff et al., 2022]. This idea still satisfies Definition 1(B)—without violating Assumptions 1-3—however, we immediately loose Definition 1(A). The reason is that breaking the symmetry by neglecting even a single ordering, $\boldsymbol{\pi} \in \mathbb{S}_n$, makes (4) inexact. To see this, let $\widehat{\mathbb{S}}_n \subset \mathbb{S}_n$ be a subset consisting of $n! - 1$ orderings, then it holds that $p(\mathbf{G}^n|n) \geq \sum_{\boldsymbol{\pi} \in \widehat{\mathbb{S}}_n} p(\mathbf{G}^n, \boldsymbol{\pi}|n)$.

## 3.4  $\mathbb{S}_n$-INVARIANCE BY SORTING

To make PGCs $\mathbb{S}_n$-invariant, we can simply sort $\mathbf{G}$ into a pre-defined canonical ordering, $\boldsymbol{\pi}_c$, each time before it enters the model as input [4]. The model then returns the same value for each permutation of $\mathbf{G}$ and is therefore $\mathbb{S}_n$-invariant. However, this design choice does not satisfy Definition 6 since we no longer operate with the exact likelihood (4) but only its lower bound. To demonstrate this, consider the variational evidence lower bound on the logarithm of (4),

$$\log p(\mathbf{G}^n|n) \geq \mathbb{E}_{q(\boldsymbol{\pi}|\mathbf{G}^n)}[\log p(\mathbf{G}, \boldsymbol{\pi}|n)] + \mathcal{H}(q(\boldsymbol{\pi}|\mathbf{G}^n)), \tag{5}$$

---

[4]Note that $\boldsymbol{\pi}_c$ is a variable (not a single fixed ordering), which sorts two graphs $\mathbf{G}_1$ and $\mathbf{G}_2$ with the same values but different orderings into their canonical representation, $\mathbf{G}_c$, i.e., we have $\boldsymbol{\pi}_c\mathbf{G}_1 = \mathbf{G}_c$ and $\boldsymbol{\pi}_c\mathbf{G}_2 = \mathbf{G}_c$ ($\boldsymbol{\pi}_c$'s are two distinct orderings). In other words, $\boldsymbol{\pi}_c$ is the output of a canonicalization algorithm whose input is $\mathbf{G}_1$ or $\mathbf{G}_2$. This algorithm provides different $\boldsymbol{\pi}_c$ for $\mathbf{G}_1$ and $\mathbf{G}_2$.

where $q(\boldsymbol{\pi}|\mathbf{G}^n)$ is the variational posterior distribution, and $\mathcal{H}$ is the differential entropy. If there is only a single canonical ordering, $\boldsymbol{\pi}_c$, then $q(\boldsymbol{\pi}|\mathbf{G}^n)$ concentrates all its mass to a single point, and (5) becomes $\log p(\mathbf{G}^n|n) \geq \log p(\mathbf{G}^n|\boldsymbol{\pi}_c, n) + c$ where $c = \log p(\boldsymbol{\pi}_c|n)$. This approach significantly reduces the computational complexity of (4) from $\mathcal{O}(n!)$ to the complexity of a selected sorting algorithm. Here, we can see that satisfying Definition 6 is uneasy. On the one hand, we reduce the factorial complexity down to the polynomial one (satisfying Definition 1(B)). On the other hand, we pay for this by involving variational approximation (sacrificing Definition 1(A)).

**Some orderings are more suitable than others.** Which of the $n!$ orderings to choose? Not all orderings are suitable. Random ordering can harm the performance of the model. However, a subset of $\mathbb{S}_n$ leads to a good performance. We refer to this collection as canonical orderings. Some rely on specific domain knowledge, whereas others are completely domain agnostic. Appendix G (Figure 6) shows an unnormalized empirical distribution over adjacency matrices for a random ordering and four canonical orderings. The ordered adjacency matrices are more structured and highly concentrated near the diagonal, which simplifies the learning of $p(\mathbf{G})$ and increases the chance of sampling semantically valid graphs (Figures 9 and 10).

**Graph sorting and the isomorphism problem.** Graph isomorphism and graph canonicalization (sorting) are two tightly connected problems. There is currently no polynomial-time solution to graph isomorphism for general graphs, though a quasi-polynomial one exists [Babai, 2016], i.e., the problem is currently considered NP-intermediate. For certain types of graphs, such as planar graphs, there are polynomial-time solutions. This characteristic was exploited to design canonicalization algorithms for molecular graphs [Faulon, 1998] (which are mostly—but not exclusively—planar). However, it was noticed that for large and/or highly symmetrical molecules, these canonicalization procedures might yield non-unique orderings. Notwithstanding that, by involving additional domain knowledge in the canonicalization process, highly reliable algorithms with tie-breaking subroutines were developed, resolving the sorting noise in such rare situations [Schneider et al., 2015]. In our experiments (Section 5), we rely on the procedure reported in [Schneider et al., 2015], which provides robust canonicalization for the datasets considered in the present paper.

**PGCs for undirected graphs.** The structural properties of $\mathbf{G}$ are characterized by the layout of $\mathbf{A}$ [5]. We consider PGCs for undirected graphs without self-loops, where $\mathbf{A}$ is uniquely determined by $N\frac{N-1}{2}n_A$ entries in its lower triangular part, $\mathbf{L}$, which allows us to avoid the repeated entries and improve the scalability. Consequently, $p(\mathbf{G}^n|n)$

---

[5]For example, if $\mathbf{A}$ is asymmetric with non-zero diagonal entries, then $\mathbf{G}$ is a directed graph with self-loops.

in (1) is the following joint probability distribution:

$$p(\mathbf{G}^n|n) \coloneqq p(\mathbf{X}^n, \mathbf{L}^n|n).$$

This PGC can be architectured in many different ways. In this paper, we focus on splitting $p(\mathbf{X}^n, \mathbf{L}^n|n)$ into two main parts, as illustrated in Figure 1. The first models the nodes, taking $\mathbf{X}^n$ as input, whereas the second models the edges, taking row-flattened $\mathbf{L}^n$ as input. The last layer of both these parts is the sum layer with $n_c$ nodes. To capture the correlations between the nodes and edges, we connect the outputs of these two parts by the product layer with $n_c$ units and then aggregate its outputs by the sum layer with a single unit. The higher the values of $n_c$, the better we capture the interactions between the nodes and edges. In Section 5, we compare different node-PC and edge-PC parts implementations.

## 4  RELATED WORK

**Intractable probabilistic models.** Practically all canonical types of DGMs have been extended to graph-structured data. These DGMs learn $p(\mathbf{G})$ (or its approximation) in different ways. Autoregressive models rely on the chain rule of probability to construct $\mathbf{G}$ by sequentially adding new nodes and edges based on the previous ones [You et al., 2018, Liao et al., 2019]. Variational autoencoders train an encoder and a decoder to map between space of graphs and latent space [Simonovsky and Komodakis, 2018, Liu et al., 2018, Ma et al., 2018, Grover et al., 2019, Kwon et al., 2019, Samanta et al., 2020]. Generative adversarial networks train (i) a generator to map from latent space to space of graphs, and (ii) a discriminator to distinguish whether the graphs are synthetic or real [De Cao and Kipf, 2018, Bojchevski et al., 2018]. Normalizing flows use the change of variables formula to transform a base distribution on latent space to a distribution on space of graphs, $p(\mathbf{G})$, using invertible neural networks [Liu et al., 2019, Luo et al., 2021]. Energy-based models define $p(\mathbf{G})$ by the Boltzmann distribution, whose energy function assigns low energies to correct graphs and high energies to incorrect graphs Liu et al. [2021]. Diffusion models systematically noise and denoise trajectories of graphs based on forward and backward diffusion processes, respectively [Jo et al., 2022, Huang et al., 2022, Vignac et al., 2023, Kong et al., 2023, Hua et al., 2024]. All these models rely on deep neural networks that contain non-linear transformations, which prevents tractable computation of probabilistic inference queries.

**Tractable probabilistic models.** TPMs for graphs in the variable-size tensor representation considered in this paper—where both the nodes and edges are attributed—have received no attention. Existing models are designed either for general (possibly cyclic) graphs without edge attributes [Zheng et al., 2018, Errica and Niepert, 2024] or for graphs of specific relational structures [Nath and Domingos, 2015, Loconte et al., 2023, Papež et al., 2024b], including trees (typically also without edge attributes) and knowledge graphs. These TPMs do not encode a fully permutation invariant probability distribution over a graph, with the only exceptions [Nath and Domingos, 2015, Papež et al., 2024b], which are permutation invariant only for certain graph sub-structures. Additionally, certain TPMs do not encode a probability distribution over a whole graph, but only its triplet relations [Loconte et al., 2023], thus proving only local tractable inference. Moreover, adopting the pseudo-likelihood in the GSPN model [Errica and Niepert, 2024] renders the inference intractable even from the standard perspective of tractability (Definition 1). Aside from [Papež et al., 2024b], these TPMs do not model the cardinality of a graph, which hinders their generative capabilities. Similarly, excluding [Loconte et al., 2023] and partially [Nath and Domingos, 2015, Papež et al., 2024b], these TPMs rely on a fixed graph ordering. We provide additional remarks on the related work in Appendix C.

## 5  EXPERIMENTS

We evaluate PGCs in the following three experiments. First, we compare $\mathbb{S}_n$-sensitive and $\mathbb{S}_n$-invariant PGCs. Second, we assess their ability to generate novel and semantically valid graphs. Third, to demonstrate their aptness for tractable probabilistic inference, we use them to generate new graphs conditionally on known graph substructures. The experiments are performed in the context of the computational design of molecular graphs. We provide the code at https://github.com/mlnpapez/PGC.

**Molecular design.** DGMs for molecular graphs are important in discovering drugs and materials with desired properties. Given a dataset of molecular graphs, the task is to learn a probability distribution of chemically valid molecules, $p(\mathbf{G})$. This is a complex combinatorial task, as not all combinations of atoms and bonds can be connected, but the connections must adhere to chemical valency rules. The aim is that $p(\mathbf{G})$ can produce molecules that were unseen during the training but satisfy the valency constraints.

**Datasets.** We test the performance of PGCs on QM9 [Ramakrishnan et al., 2014] and ZINC250k [Irwin et al., 2012] datasets, which are two standard benchmarks that are often used to assess DGMs for molecular design. We summarize the statistics of these datasets in Table 2 and describe their preprocessing in Appendix F.1.

**Metrics.** We evaluate the percentage of *valid*, *unique*, and *novel* molecules. Valid molecules are those that satisfy chemical valency rules (without resorting to any corrections mechanisms). Unique molecules are valid molecules that are not a duplicate of other generated molecules. Novel molecules are valid and unique molecules that are not in the training data. We further compute the Fréchet ChemNet distance (FCD) [Preuer et al., 2018], which is the distance between

Table 1: *Unconditional generation on the QM9 and Zinc250k datasets.* The mean value and standard deviation of the molecular metrics for the baseline intractable DGMs (top) and various implementations of the $\pi$PGCs that rely on the sorting to ensure the $\mathbb{S}_n$-invariance (bottom). The results are computed over five runs with different initialization. The 1st, 2nd, and 3rd best results are highlighted in colors.

| Model | QM9 | | | | | Zinc250k | | | | |
|---|---|---|---|---|---|---|---|---|---|---|
| | Valid↑ | NSPDK↓ | FCD↓ | Unique↑ | Novel↑ | Valid↑ | NSPDK↓ | FCD↓ | Unique↑ | Novel↑ |
| GraphAF | 74.43±2.55 | 0.021±0.003 | 5.27±0.40 | 88.64±2.37 | 86.59±1.95 | 68.47±0.99 | 0.044±0.005 | 16.02±0.48 | 98.64±0.69 | 100.00±0.00 |
| GraphDF | 93.88±4.76 | 0.064±0.000 | 10.93±0.04 | 98.58±0.25 | 98.54±0.48 | 90.61±4.30 | 0.177±0.001 | 33.55±0.16 | 99.63±0.01 | 99.99±0.01 |
| MoFlow | 91.36±1.23 | 0.017±0.003 | 4.47±0.60 | 98.65±0.57 | 94.72±0.77 | 63.11±5.17 | 0.046±0.002 | 20.93±0.18 | 99.99±0.01 | 100.00±0.00 |
| EDP-GNN | 47.52±3.60 | 0.005±0.001 | 2.68±0.22 | 99.25±0.05 | 86.58±1.85 | 82.97±2.73 | 0.049±0.006 | 16.74±1.30 | 99.79±0.08 | 100.00±0.00 |
| GraphEBM | 8.22±2.24 | 0.030±0.004 | 6.14±0.41 | 97.90±0.14 | 97.01±0.17 | 5.29±3.83 | 0.212±0.005 | 35.47±5.33 | 98.79±0.15 | 100.00±0.00 |
| SPECTRE | 87.30±n/a | 0.163±n/a | 47.96±n/a | 35.70±n/a | 97.28±n/a | 90.20±n/a | 0.109±n/a | 18.44±n/a | 67.05±n/a | 100.00±n/a |
| GDSS | 95.72±1.94 | 0.003±0.000 | 2.90±0.28 | 98.46±0.61 | 86.27±2.29 | 97.01±0.77 | 0.019±0.001 | 14.66±0.68 | 99.64±0.13 | 100.00±0.00 |
| DiGress | 99.00±0.10 | 0.005±n/a | 0.36±n/a | 96.20±n/a | 33.40±n/a | 91.02±n/a | 0.082±n/a | 23.06±n/a | 81.23±n/a | 100.00±n/a |
| GRAPHARM | 90.25±n/a | 0.002±n/a | 1.22±n/a | 95.62±n/a | 70.39±n/a | 88.23±n/a | 0.055±n/a | 16.26±n/a | 99.46±n/a | 100.00±n/a |
| BT | 78.15±0.70 | 0.004±0.001 | 1.68±0.13 | 99.66±0.10 | 93.20±0.36 | 17.00±0.55 | 0.050±0.002 | 9.42±0.36 | 100.00±0.00 | 100.00±0.00 |
| LT | 62.81±2.69 | 0.007±0.001 | 2.57±0.24 | 99.72±0.05 | 95.95±0.35 | 4.11±0.24 | 0.056±0.001 | 11.53±0.19 | 100.00±0.00 | 100.00±0.00 |
| RT | 81.83±1.62 | 0.003±0.000 | 1.29±0.04 | 99.37±0.07 | 90.67±0.40 | 6.90±0.58 | 0.047±0.000 | 9.59±0.39 | 100.00±0.00 | 100.00±0.00 |
| RT-S | 88.83±0.75 | 0.002±0.000 | 1.11±0.01 | 99.38±0.06 | 88.49±0.45 | 14.66±0.66 | 0.043±0.002 | 8.78±0.34 | 100.00±0.00 | 100.00±0.00 |
| HCLT | 89.00±0.17 | 0.003±0.000 | 1.45±0.09 | 99.45±0.04 | 90.62±0.43 | 23.67±0.45 | 0.035±0.000 | 8.93±0.06 | 100.00±0.00 | 100.00±0.00 |

the generated and training molecules computed from the activations of the second last layer of ChemNet; and the neighborhood subgraph pairwise distance kernel (NSPDK) MMD [Costa and Grave, 2010], which is the distance between the generated and test molecules considering the node and edge features.

**PGC variants.** We distinguish PGCs with the following $\mathbb{S}_n$-invariance principles: inherent $\mathbb{S}_n$-invariance (Section 3.2), $\mathbb{S}_n$-invariance by marginalization (Section 3.3), $\mathbb{S}_n$-invariance by conditioning on a canonical ordering, $p(\mathbf{G}|\boldsymbol{\pi}_c)$, (Section 3.4), and no $\mathbb{S}_n$-invariance, which we refer to as $i$PGC, $n!$PGC, $\pi$PGC, and $s$PGC, respectively. All these models follow the design template in Figure 1.

**$\mathbb{S}_n$-sensitivity of PGCs.** A properly designed graph DGM should be able to recognize a single instance of a graph, $\mathbf{g}^n$, up to all its $n!$ permutations. This is especially important in anomaly detection since we do not want our model to detect different graph permutations as anomalous. Therefore, we compare the $\mathbb{S}_n$-invariant PGCs from Section 3 with an $\mathbb{S}_n$-sensitive PGC on a synthetic anomaly detection example. To carry out the experiment in a feasible time span (due to the $n!$ complexity of (4)), we restrict ourselves to only a subset of the QM9 dataset containing molecules with up to 6 atoms. We train the models on molecules with up to 5 atoms (in-distribution data) and use the molecules with 6 atoms as the anomalies (out-of-distribution data), making these two subsets perfectly balanced in their size. However, we randomly permute 20% of the in-distribution molecules during the evaluation. Figure 2 draws the histogram of the exact log-likelihood $\log p(\mathbf{G})$ or its lower bound $\log p(\mathbf{G}|\boldsymbol{\pi}_c)$, and the area under the curve (AUC) obtained by computing the true positive rate (TPR) and the true negative rate (TNR). All the $\mathbb{S}_n$-invariant PGCs correctly recognize the permuted molecules as in-distribution data. In contrast, the $\mathbb{S}_n$-sensitive PGC incorrectly recognizes them as out-of-distribution data by assigning them very low log-likelihoods. This behavior has a severe impact on the AUC metric. While the $\mathbb{S}_n$-invariant PGCs achieve near-perfect AUC, the $\mathbb{S}_n$-sensitive PGC suffers from non-zero FPR.

In this experiment, the number of parameters in $i$PGC is 10x higher than in $\pi$PGC, $n!$PGC, or $s$PGC. Still, we observe in Figure 2 that $\pi$PGC consistently outperforms $i$PGC, confirming our previous claim about the reduced expressive power of $i$PGCs (Section 3). Therefore, we continue our experiments only with $\pi$PGC, leaving $n!$PGC and $s$PGC due to their factorial complexity and $\mathbb{S}_n$-sensitivity, respectively.

**$\pi$PGC variants.** The PGCs (Section 3) can be instantiated in different ways, depending on the region graph (RG) [Dennis and Ventura, 2012] that is used to build the architecture of the node-PC and edge-PC (Figure 1). We consider the following types of region graphs: binary tree (BT) [Loconte et al., 2024b], linear tree (LT) [Loconte et al., 2024a], randomized tree (RT) [Peharz et al., 2020a,b], randomized tree with synchronized permutations between the node and edge PCs (RT-S), and the hidden Chow-Liu tree (HCLT) [Liu and Van den Broeck, 2021]. Note that while the first four variants do not rely on data to build the RG, the last one learns the structure of the Chow-Liu tree from data. We consider that all these RGs are used to build a tensorized, monotonic architecture (Definition 12). We perform an extensive gridsearch over the hyper-parameters of these architectures, as detailed in Appendix F. The ordering is also a hyper-parameter and covers all cases discussed in Appendix G.

**Baselines.** We compare $\pi$PGCs with the following intractable DGMs: GraphAF [Shi et al., 2020], GraphDF [Luo et al., 2021], MoFlow [Zang and Wang, 2020], EDP-GNN [Niu et al., 2020], GraphEBM [Liu et al., 2021], SPECTRE [Martinkus et al., 2022], GDSS [Jo et al., 2022], DiGress [Vignac et al., 2023], GraphARM [Kong et al., 2023]. Additional details about these models are given in Appendix F.4.

**Unconditional molecule generation.** Table 1 shows that all the $\pi$PGCs variants deliver a competitive performance across all metrics on the QM9 dataset. Most notably, the RT-S variant scores 1st in NSPDK and 2nd in FCD. For the Zinc250k dataset, the RT-S model is 3rd in NSPDK and 1st in FCD, whereas the HCLT model is 2nd in both these metrics. However, the validity drops for the Zinc250k dataset, where even the best $\pi$PGC variant (HCLT) delivers

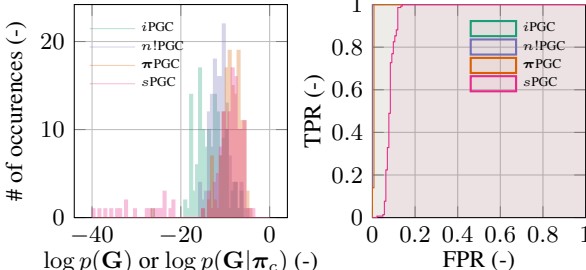

Figure 2: $\mathbb{S}_n$-*sensitivity of PGCs.* The histograms of $\log p(\mathbf{G})$ and $\log p(\mathbf{G}|\boldsymbol{\pi}_c)$ (left) and the AUC (right) for different $\mathbb{S}_n$-invariance mechanisms of PGCs in the context of anomaly detection.

only 23% validity. Recall that we do not use any validity corrections mechanism [Zang and Wang, 2020] to satisfy the valency rules but rely on our model to learn them on its own. The validity metric is considered unreliable, as it can be artificially increased by generating simpler molecules even with rule-based systems [Preuer et al., 2018]. The NSPDK and FCD metrics are more robust, revealing better diversity of the generated molecules and similarity to real molecules.

**Conditional molecule generation.** To illustrate that PGCs can answer probabilistic inference queries, Figure 3 displays conditional molecule generation for the RT-S variant, where we can see that each molecule's newly generated part varies in size and composition. More examples of the conditional generation, along with the corresponding molecular metrics, are provided in Appendix H, which also contains a detailed comparison of all $\boldsymbol{\pi}$PGC variants for different orderings in the context of the unconditional generation.

# 6   CONCLUSION AND LIMITATIONS

**Conclusion.** We have proposed PGCs—tractable probabilistic models for graphs—and specified the conditions for their tractability through the extended definition of $\mathbb{S}_n$-invariance. Furthermore, we have developed marginalization padding, which conveniently utilizes the tractability of PGCs to handle the variable-size nature of graphs. We have shown that designing PGCs with the inherently build-in, tractable $\mathbb{S}_n$-invariance—i.e., without using any specific $\mathbb{S}_n$-invariance mechanisms—requires rather strict conditional i.i.d. assumptions on modeling the nodes and edges, which undermines the expressive power of such PGCs. Therefore, we have introduced PGCs that are $\mathbb{S}_n$-invariant through the sorting. Despite not complying with the strict form of tractable $\mathbb{S}_n$-invariance, they answer the same inference queries as conventional PCs. Importantly, these PGCs outperform, or are competitive with, existing intractable DGMs in most standard molecular metrics except for the validity on the Zinc250k dataset.

**Limitations.** The low validity on the Zinc250k dataset can easily be improved by rejection sampling, which is computationally cheap with PGCs, as they produce samples in a single pass through the network. This contrasts with au-

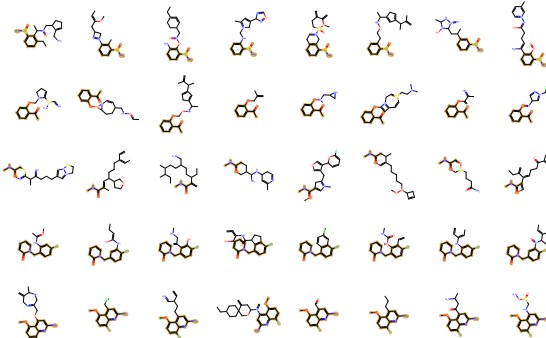

Figure 3: *Conditional generation on the Zinc250k dataset.* The yellow area highlights the known part of the molecule. There is one such known part per row. Each column corresponds to a new molecule generated conditionally on the known part.

toregressive and diffusion models, where high validity is substantially more important since they generate a single sample iteratively, making several expensive passes through the model. We conjecture that the low validity comes from the template in Figure 1, where the node and edge PCs are connected only through a mixture of independent components. This architecture captures the correlations between the nodes and edges by a sufficiently high number of components. However, it can be improved by connecting the node and edge PCs also between their lower layers. To make this enhancement, we would require PCs with an input layer that allows for hybrid variables (as there are different numbers of categories for the nodes and edges), which is currently unavailable with the existing PC libraries. Moreover, as for other approaches relying on dense graph representations, our current instance of PGCs scales with $\mathcal{O}(m^2)$ complexity, making it less suitable for large-scale graphs. In future work, we will address these limitations by proposing new instances of PGCs.

### Acknowledgements

The authors acknowledge the support of the GAČR grant no. GA22-32620S and the OP VVV funded project CZ.02.1.01/0.0/0.0/16_019/0000765 "Research Center for Informatics".

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

# Probabilistic Graph Circuits: Deep Generative Models for Tractable Probabilistic Inference over Graphs
## (Supplementary Material)

**Milan Papež**[1]    **Martin Rektoris**[1]    **Václav Šmídl**[1]    **Tomáš Pevný**[1]

[1]Artificial Intelligence Center, Czech Technical University, Prague, Czech Republic

## A    PERMUTATION INVARIANCE

$\mathbb{S}_n$**-invariance.** Exchangeable data structures, including sets, graphs, partitions, and arrays [Orbanz and Roy, 2014], have a factorial number of possible configurations (orderings). Permutation invariance says that no matter a selected configuration, the probability of an exchangeable data structure has to remain the same.

To define the permutation invariance of a probability distribution, let $\mathbb{S}_n$ be a finite symmetric group of a set of $n$ elements. This is a set of all $n!$ permutations of $[n] \coloneqq (1, \ldots, n)$, where any of its members, $\boldsymbol{\pi} \in \mathbb{S}_n$, permutes an $n$-dimensional vector, $\mathbf{X} \coloneqq (X_1, \ldots, X_n)$, as follows: $\boldsymbol{\pi}\mathbf{X} = (X_{\pi(1)}, \ldots, X_{\pi(n)})$.

**Definition 8.** ($\mathbb{S}_n$-invariance of vectors). The probability distribution $p$ is permutation invariant iff $p(\mathbf{X}) = p(\boldsymbol{\pi}\mathbf{X})$ for all $\boldsymbol{\pi} \in \mathbb{S}_n$. $\mathbf{X}$ is permutation invariant if $p(\mathbf{X})$ is.

## B    CIRCUITS

In this section, to simplify the notation, we consider that the size of $\mathbf{X}$ is fixed, $N = n$. Therefore, we leave the explicit dependence on $n$ out of $p(\mathbf{X}^n|n)$.

**Definition 9** (Circuit [Vergari et al., 2021]). A *circuit* $c$ over random variables $\mathbf{X} \coloneqq \{X_1, \ldots, X_n\}$ is a parameterized, directed, acyclic computational graph encoding a function $c(\mathbf{X})$. It contains three types of computational *units*: *input*, *product*, and *sum*. All sum and product units receive the outputs of other units as inputs. We denote the set of inputs of a unit $u$ as $\mathsf{in}(u)$. Each unit $u$ encodes a function $c_u$ over a subset of the random variables, $\mathbf{X}_u \subseteq \mathbf{X}$, which we refer to as *scope*. The input unit $c_u(\mathbf{X}_u) \coloneqq f_u(\mathbf{X}_u)$ computes a user-defined parameterized function, $f_u$; the sum unit computes the weighted sum of its inputs, $c_u(\mathbf{X}_u) \coloneqq \sum_{i \in \mathsf{in}(u)} w_i c_i(\mathbf{X}_i)$, where $w_i \in \mathbb{R}$ are the weight parameters; and the product unit computes the product of its inputs, $c_u(\mathbf{X}_u) \coloneqq \prod_{i \in \mathsf{in}(u)} c_i(\mathbf{X}_i)$. The scope of any sum or product unit is the union of its input scopes, $\mathbf{X}_u = \bigcup_{i \in \mathsf{in}(u)} \mathbf{X}_i$. A circuit can have one or multiple root units. The scope of a root unit is $\mathbf{X}$.

**Definition 10** (Probabilistic circuit). A *probabilistic circuit* (PC) over random variables $\mathbf{X}$ is a circuit (Definition 9) $c$, such that $\forall \mathbf{x} \in \mathsf{dom}(\mathbf{X}) : c(\mathbf{x}) \geq 0$, i.e., it is a *non-negative* function for all values of $\mathbf{X}$.

A PC (Definition 10) can encode a possibly unnormalized probability distribution, $p(\mathbf{x}) \propto c(\mathbf{x})$.

**Tractable probabilistic inference.** PCs are *tractable* if they provide *exact* and *efficient* answers to inference queries over arbitrary subsets of $\mathbf{X}$ [Choi et al., 2020, Vergari et al., 2021]. Here, exact means that the answers do not involve any approximations, and efficient means that the answers are computed in polytime. An arbitrary inference query can be expressed in terms of the expectation $\mathbb{E}_{p(\mathbf{X})}[h(\mathbf{X})] = \int h(\mathbf{X})p(\mathbf{X})d\mathbf{X}$, where $h$ is a function that allows us to formulate a desired query over $\mathbf{X}$ or its part(s). However, to make the computation of $\mathbb{E}_p[h]$ tractable, both $p$ and $h$ have to satisfy certain assumptions.

**Assumption 1** (Smoothness [Darwiche and Marquis, 2002]). A circuit $c$ is *smooth* if the inputs of each sum unit $u \in c$ have the same scope, $\forall a, b \in \mathsf{in}(u) : \mathbf{X}_a = \mathbf{X}_b$.

**Assumption 2** (Decomposability [Darwiche and Marquis, 2002]). A circuit $c$ is *decomposable* if the inputs of each product unit $u \in c$ have pair-wise disjoint scopes $\forall a, b \in \text{in}(u) : \mathbf{X}_a \cap \mathbf{X}_b = \varnothing$.

**Assumption 3** (Tractable input units). A circuit $c$ has *tractable input units* if each input unit $u \in c$ admits an algebraically closed-form solution to its integral, $\int_{\text{dom}(\mathbf{Z}_u)} c_u(\mathbf{y}_u, \mathbf{z}_u) d\mathbf{Z}_u$, for any $\mathbf{X}_u \coloneqq \{\mathbf{Y}_u, \mathbf{Z}_u\}$.

To satisfy Assumption 3, we need each $c_u(\mathbf{X}_u)$ to belong to a tractable family of probability distributions [Barndorff-Nielsen, 1978].

**Assumption 4** (Compatibility [Vergari et al., 2021]). Two circuits $f$ and $g$ over the same random variables $\mathbf{X}$ are *compatible* if they satisfy Assumptions 1 and 2, and any two product units $a \in f$ and $b \in g$, such that $\mathbf{X}_a = \mathbf{X}_b$, can be rearranged into pair-wise compatible products that decompose in the same way: $(\mathbf{X}_a = \mathbf{X}_b) \Rightarrow (\mathbf{X}_{a_i} = \mathbf{X}_{b_i}, a_i \text{ and } b_i \text{ are compatible})$ for some rearrangement of the inputs of $a$ (resp. $b$) into $a_1$, $a_2$ (resp. $b_1$, $b_2$).

**Definition 11** (Tractable expectation). Two circuits $f$ and $g$ over the same random variables $\mathbf{X}$ admit tractable expectation, $\mathbb{E}_g[f] \coloneqq \int f(\mathbf{X})g(\mathbf{X})d\mathbf{X}$, if they satisfy Assumption 4, and the product of any two input units $a \in f$ and $b \in g$, such that $\mathbf{X}_a = \mathbf{X}_b$, can be integrated tractably.

**Definition 12** (Tensorized PCs). A tensorized PC [Peharz et al., 2020a,b, Loconte et al., 2024b] is a deep learning model of a probability distribution, $p(\mathbf{X})$, over a *fixed-size* vector, $\mathbf{X} \in \mathcal{X}$. The network contains several layers of computational units (similar to neural networks [Vergari et al., 2019b]). Each layer is defined over its *scope*, $\mathbf{X}_u \subseteq \mathbf{X}$, i.e., a subset of the input. There are three types of layers, depending on the units they encapsulate: sum layer $\mathsf{L_S}$, product layer $\mathsf{L_P}$, and input layer $\mathsf{L_I}$. The units of *input* layers are user-defined probability distributions, $p_{u,i}(\mathbf{X}_u)$. For $n_I$ units, an input layer computes $p_{u,i}(\mathbf{X}_u)$ for $i \in (1, \ldots, n_I)$ and outputs an $n_I$-dimensional vector of probabilities $\mathbf{l}$. The units of *product* layers are factored distributions, applying conditional independence over a pair-wise disjoint partition of their scope. A product layer receives outputs from $n$ layers, $(\mathbf{l}_1, \ldots, \mathbf{l}_n)$, and computes either an Hadamard product, $\mathbf{l} = \odot_{i=1}^n \mathbf{l}_i$, or Kronecker product, $\mathbf{l} = \otimes_{i=1}^n \mathbf{l}_i$. The units of *sum* layers are mixture distributions. For $n_S$ units, a sum layer receives an $n$-dimensional input, $\mathbf{l}$, from a previous layer and computes $\mathbf{W}\mathbf{l}$, where $\mathbf{W}$ is an $n_S \times n$ matrix of row-normalized weights. The output (layer) of a tensorized PC is typically a sum layer.

**PCs can only be partially $\mathbb{S}_n$-invariant.** The study of $\mathbb{S}_n$-invariance of PCs has received only limited attention until recently [Papež et al., 2024b], where the authors prove that even if the input units of a PC are permutation invariant, then the traditional structural restrictions—i.e., smoothness (Assumption 1) and decomposability (Assumption 2)—make this PC only partially permutation invariant. For this reason, we consider conventional PCs to be permutation-sensitive distributions in the present paper.

## C  ADDITIONAL NOTES ON RELATED WORK

In our previous work [Papež et al., 2024a], we investigated different approaches for achieving the $\mathbb{S}_n$-invariance with PCs, concluding that sorting achieves the best performance in unconditional molecule generation. The present paper provides several key advances over [Papež et al., 2024a], which are detailed in the last paragraph of Section 1. In particular, one important difference is that [Papež et al., 2024a] relies on zero padding to handle the variable-size character of graphs. In contrast, we use the marginalization padding (Section 3) in the current paper. In our experiments, we observed that the zero padding delivers performance similar to the marginalization padding in terms of the metrics presented in Table 1. However, these two approaches are principally incomparable in terms of the likelihood. Indeed, the marginalization padding reflects the size of a graph in its likelihood, while the zero padding does not. This phenomenon can seriously undermine the effectiveness of zero padding in the context of anomaly detection. We will investigate this further in future work.

## D  TRACTABLE INFERENCE QUERIES OVER GRAPHS

Probabilistic inference queries over graphs—such as marginalization, conditioning, and expectation—are important in various applications. The examples include explainable anomaly detection (marginalization Ying et al. [2019]), retro-synthesis (conditioning Igashov et al. [2024]), and uncertainty quantification (expectation Yang and Li [2023]). However, many graph DGMs are intractable and thus do not permit even the most basic inference queries. On the contrary, PGCs are tractable DGMs that allow for a broad range of inference queries.

$$h(\mathbf{G}_a^3, \mathbf{G}_b^1; 4) = \prod_{t \notin \mathbf{s}} \mathbb{1}_{\mathbf{x}_t}(\mathbf{X}_t) \prod_{u \notin \mathbf{s}} \mathbb{1}_{\mathbf{a}_{tu}}(\mathbf{A}_{tu}) \mathbb{1}_{\mathrm{dom}(\mathbf{X}_2)}(\mathbf{X}_2)$$
$$\times \mathbb{1}_{\mathrm{dom}(\mathbf{A}_{22})}(\mathbf{A}_{22}) \prod_{k \notin \mathbf{s}} \mathbb{1}_{\mathrm{dom}(\mathbf{A}_{2k})}(\mathbf{A}_{2k}) \mathbb{1}_{\mathrm{dom}(\mathbf{A}_{k2})}(\mathbf{A}_{k2})$$

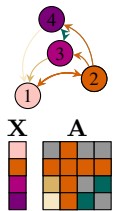 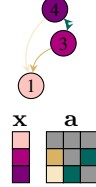

(a) An example of (7) for $\mathbf{s} := \{2\}$  (b) The corresponding node of $\mathbf{G}$  (c) $\mathbf{G}$ after the marginal evidence query

Figure 4: *An example of the marginal evidence query over a 4-node graph.* (a) An instantiation of the omni-compatible GC (7) over a graph $\mathbf{G}^4$ with targetting the 2nd node, $\mathbf{s} := \{2\}$. The orange color highlights the targeted node and its associated edges. (b) A visual representation of $\mathbf{G}$, where the targeted node and its associated edges, which correspond to (a), are highlighted in orange. (c) After performing the marginal evidence query over the 2nd node of $\mathbf{G}^4$, we obtain a new 3-node evidence graph $\mathbf{g}$.

**Tractable expectation.** Most inference queries can collectively be expressed in terms of the following expectation:

$$\mathbb{E}_{p(\mathbf{G})}[h(\mathbf{G})] = \sum_{n=1}^{\infty} \int h(\mathbf{g}^n) p(\mathbf{g}^n | n) p(n) d\mathbf{g}^n, \tag{6}$$

where $h$ is a GC (Definition 4) whose specific form allows us to formulate a desired query over (a part of) $\mathbf{G}$.

Before focusing on specific instances of (6), we introduce the conditions for its tractability in Proposition 4.

**Proposition 4.** *(Tractable expectation of a GC.) Let $h$ and $p$ be two tractable (Proposition 1) and compatible (Assumption 4) (P)GCs, such that the product of any two input units $u \in h$ and $v \in p$ with the same scope $\mathbf{G}_u = \mathbf{G}_v$ (Definition 3) has a tractable integral; then, the expectation (6) is tractable.*

Apart from Proposition 1, the key requirement for the tractability of (6) is the *compatibility* of $h$ and $p$. The standard definition of compatibility (Assumption 4) does not change under our variable-size (C2) and $\mathbb{S}_n$-invariant (C3) setting since $\mathbf{g}^n$ enters both $h$ and $p$ as the same realization (with a fixed size and values). Note that the part of Proposition 1 requiring the restricted support of the cardinality distribution can be relaxed for $h$, as it is satisfied by $p$.

**Querying $k$-node subgraphs.** There can be many examples of a GC, $h$, that is compatible with a PGC, $p$. We will focus on a simple, yet very powerful, omni-compatible Vergari et al. [2021], and tractably $\mathbb{S}_n$-invariant circuit, which will allow us to formulate many desired inference queries.

**Proposition 5.** *(Omni-Compatible Tractably $\mathbb{S}_n$-invariant GC.) Let $\mathbf{G}^n = \{\mathbf{G}_a^{n-k}, \mathbf{G}_b^k\}$, where $\mathbf{G}_a^{n-k}$ and $\mathbf{G}_b^k$ are connected $(n-k)$-node and $k$-node subgraphs of $\mathbf{G}^n$. A fully factorized, $n$-dependent, GC of the following form:*

$$h(\mathbf{G}_a^{n-k}, \mathbf{G}_b^k; n) := \prod_{t \notin \mathbf{s}} h(\mathbf{X}_t) \prod_{u \notin \mathbf{s}} h(\mathbf{A}_{tu})$$
$$\times \prod_{i \in \mathbf{s}} h(\mathbf{X}_i) \prod_{j \in \mathbf{s}} h(\mathbf{A}_{ij}) \prod_{l \notin \mathbf{s}} h(\mathbf{A}_{il}) h(\mathbf{A}_{li}) \tag{7}$$

*is tractably $\mathbb{S}_n$-invariant and omni-compatible. $\mathbf{s} := \{s_1, \ldots, s_k\} \subseteq [n]$ are indices of $k$ nodes of $\mathbf{G}^n$ belonging to $\mathbf{G}_b^k$.*

The omni-compatible GC (7) is simply the product of functions over all node and edge features in $\mathbf{G}^n$, which is rearranged into two sets $\mathbf{G}_a^{n-k}$ and $\mathbf{G}_b^k$ (Appendix E.2).

**Targeting nodes and (or) edges.** The utility of Proposition 5 lies in that a suitable choice of $h(\mathbf{X}_i)$ and $h(\mathbf{A}_{ij})$ allows us to target $i$-th node of $\mathbf{G}$ and the edge between $i$-th and $j$-th node of $\mathbf{G}$, respectively. These functions facilitate the formulation of various queries of interest. For example, if $h(\mathbf{X}_i) := \mathbb{1}_{\mathcal{X}}(\mathbf{X}_i)$, where $\mathbb{1}_{\mathcal{X}}$ is the indicator function, and $h(\mathbf{A}_{ij}) := \mathbb{1}_{\mathcal{A}}(\mathbf{A}_{ij})$, we can obtain basic inference queries just by a mere choice of the sets $\mathcal{X}$ and $\mathcal{A}$. The *evidence* query is computed for $\mathcal{X} := \mathbf{x}_i \in \mathrm{dom}(\mathbf{X}_i)$ and $\mathcal{A} := \mathbf{a}_{ij} \in \mathrm{dom}(\mathbf{A}_{ij})$. The *marginal* query is given by $\mathcal{X} := \mathrm{dom}(\mathbf{X}_i)$ and $\mathcal{A} := \mathrm{dom}(\mathbf{A}_{ij})$. $r$-th *moment* query results from $h(\mathbf{X}_i) := \mathbf{X}_i^{r_i}$ and $h(\mathbf{A}_{ij}) := \mathbf{A}_{ij}^{r_{ij}}$, where $r_i$ and $r_{ij}$ are non-negative integers. We demonstrate an example of the marginal query for $\mathbf{s} := \{2\}$ in Figure 4.

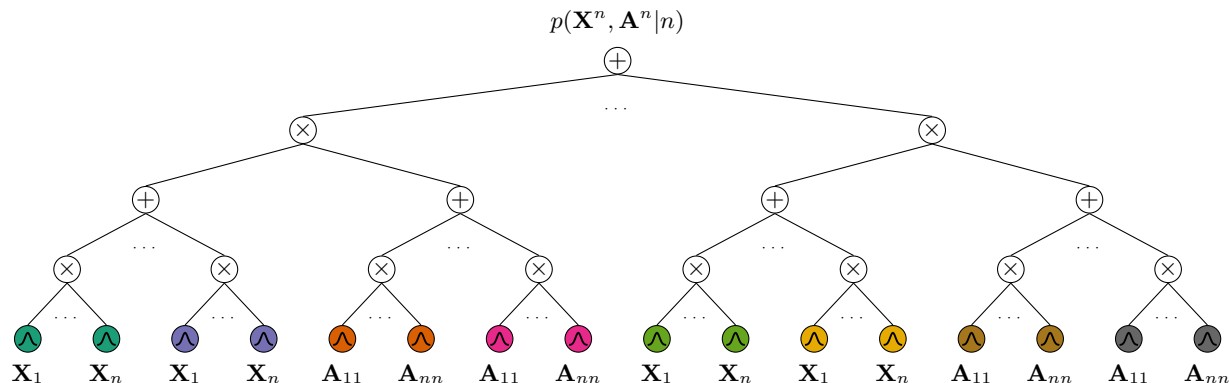

$p(\mathbf{X}^n, \mathbf{A}^n | n)$

Figure 5: *An inherently $\mathbb{S}_n$-invariant PGC through the conditional i.i.d. assumption.* The $n$-conditioned part of a PGC that is tractable and inherently $\mathbb{S}_n$-invariant, as formulated in Proposition 3. The input units with the same color share the parameterization and correspond to the product terms in (3).

# E  PROOFS

## E.1  PROOF OF PROPOSITION 3

To prove that the PGC from Proposition 3 is inherently $\mathbb{S}_n$-invariant, we need to show that the components (3) are $\mathbb{S}_n$-invariant according to the definition of the tractable $\mathbb{S}_n$-invariance in Definition 6. The i.i.d. assumption means that the parameters of the product terms in 3 are identical. To make the explicit, let us write

$$p(\mathbf{G}^n | \mathbf{z}, n) = \prod_{i \in [n]} p(\mathbf{X}_i | \mathbf{z}, n; \theta_X) \prod_{j \in [n]} p(\mathbf{A}_{ij} | \mathbf{z}, n; \theta_A),$$

where $\theta_X$ and $\theta_A$ are parameters associated with the nodes and edges, respectively. It is easy to see now that

$$\prod_{i \in [n]} p(\mathbf{X}_{\pi(i)} | \mathbf{z}, n; \theta_X) \prod_{j \in [n]} p(\mathbf{A}_{\pi(i)\pi(j)} | \mathbf{z}, n; \theta_A) = \prod_{i \in [n]} p(\mathbf{X}_i | \mathbf{z}, n; \theta_X) \prod_{j \in [n]} p(\mathbf{A}_{ij} | \mathbf{z}, n; \theta_A),$$

for all $\boldsymbol{\pi} \in \mathbb{S}_n$. Since this invariance also holds for all $\mathbf{z} \in \operatorname{dom}(\mathbf{Z})$ and propagates through the summation in (2), we can conclude that the PGC from Proposition 3 is inherently $\mathbb{S}_n$-invariant. □

## E.2  PROOF OF PROPOSITION 5

To demonstrate that $h(\mathbf{G}^n; n)$ is inherently $\mathbb{S}_n$-invariant, let us first make the parameterization of the individual terms $h(\mathbf{X}_i)$ and $h(\mathbf{A}_{ij})$ explicit as follows:

$$h(\mathbf{G}^n; n) := \prod_{i \in [n]} h(\mathbf{X}_i; \theta_i) \prod_{j \in [n]} h(\mathbf{A}_{ij}; \theta_{ij}),$$

where $\theta_i$ and $\theta_{ij}$ are parameters associated to $\mathbf{X}_i$ and $\mathbf{A}_{ij}$, respectively. It holds that $h(\boldsymbol{\pi}\mathbf{G}^n; n) = h(\mathbf{G}^n; n)$, for all $\boldsymbol{\pi} \in \mathbb{S}_n$, only if $\theta_i := \theta_X$ and $\theta_{ij} := \theta_A$ for all $i \in [n]$ and $j \in [n]$,

$$\prod_{i \in [n]} h(\mathbf{X}_{\pi(i)}; \theta_X) \prod_{j \in [n]} h(\mathbf{A}_{\pi(i)\pi(j)}; \theta_A) = \prod_{i \in [n]} h(\mathbf{X}_i; \theta_X) \prod_{j \in [n]} h(\mathbf{A}_{ij}; \theta_A).$$

This fully factorized form is what allows us to write $h(\mathbf{G}^n; n) = h(\mathbf{G}_a^{n-k}, \mathbf{G}_b^k; n) := h(\mathbf{G}_a^{n-k}; n-k) h(\mathbf{G}_b^k; k)$, where

$$h(\mathbf{G}_a^{n-k}; n-k) = \prod_{t \notin \mathbf{s}} h(\mathbf{X}_t; \theta_X) \prod_{u \notin \mathbf{s}} h(\mathbf{A}_{tu}; \theta_A),$$

$$h(\mathbf{G}_b^k; k) = \prod_{i \in \mathbf{s}} h(\mathbf{X}_i; \theta_X) \prod_{j \in \mathbf{s}} h(\mathbf{A}_{ij}; \theta_A) \prod_{l \notin \mathbf{s}} h(\mathbf{A}_{il}; \theta_A) h(\mathbf{A}_{li}; \theta_A).$$

Here, we can see that $\mathbf{G}_a^{n-k} := \{\mathbf{X}_t, \mathbf{A}_{tu}\}_{u,t\notin\mathbf{s}}$ and $\mathbf{G}_b^k := \{\mathbf{X}_i, \mathbf{A}_{ij}\}_{i,j\in\mathbf{s}} \cup \{\mathbf{A}_{il}, \mathbf{A}_{li}\}_{i,l\notin\mathbf{s}}$.

The tractability of this $\mathbb{S}_n$-invariant function follows from the fact that it does not yield an approximate value for any of the $n!$ configurations of $\mathbf{G}^n$ (satisfying Definition 1(A)), and from its linear complexity (satisfying Definition 1(B)).

A fully decomposable GC, $h(\mathbf{G}^n; n)$, is omni-compatible if it is compatible (Assumption 4) with any tractably $\mathbb{S}_n$-invariant (Definition 6) GC over $\mathbf{G}^n$. $\qquad\square$

# F  EXPERIMENTAL DETAILS

The results for the baselines in Table 1(top) are obtained from [Jo et al., 2022, Kong et al., 2023].

## F.1  DATASETS

QM9 contains around 134k stable small organic molecules of at most 9 atoms that can take 4 different types. ZINC250k contains around 250k drug-like molecules of at most 38 atoms that can take 9 different types. We use the RDKit library [Landrum et al., 2006] to first kekulize the molecules and then remove the hydrogen atoms. The final molecules contain only the single, double, and triple bonds. Since our objective is to test different canonical graph orderings, we randomly permute the atoms in each molecule before applying the canonicalization (sorting) in order let each ordering result from the same initial graph configuration. We randomly partition the datasets into 80%, 10%, and 10% for train, validation, and testing, respectively. We repeat all experiments for 5 different seeds, i.e., 5 different train, validation, and testing splits.

Table 2: *Statistics of the molecular datasets.* The meaning of the symbols is as follows: $I$ is the number of instances, $N$ is the number of atoms in a single molecule, $n_X$ is the number of atom types, $n_A$ is the number of bond types, including the empty bond.

| Dataset | $I$ | min $N$ | max $N$ | $n_X$ | $n_A$ |
|---|---|---|---|---|---|
| QM9 | 133,885 | 1 | 9 | 4 | 3+1 |
| Zinc250k | 249,455 | 6 | 38 | 9 | 3+1 |

## F.2  TRAINING SETUP

We train all $\pi$PGCs variants by minimizing the negative log-likelihood for 40 epochs. We use the ADAM optimizer [Kingma and Ba, 2014] with 256 samples in the minibatch, step-size $\alpha = 0.05$, and decay rates $\beta_1 = 0.9$ and $\beta_2 = 0.82$. All experiments are repeated 5 times with different initialization of the model's parameters. Following the baselines, we sample 10000 molecules to compute the metrics introduced in Section 5.

## F.3  PGC VARIANTS

The variants of PGCs described in Section 5 are instantiated depending on a concrete form of the region graph that is used to build the node and edge PCs (Figure 1). The expressive power of these monolithic and tensorized variants of PCs is driven by different sets of hyper-parameters, which we summarize in Table 3. All these PGCs use the input units as categorical distributions, and their cardinality distribution is also the categorical distribution (whose parameters are trained via gradient descent along with all the parameters of the node and edge PCs). We run each combination of the hyper-parameters in Table 3 for each canonical ordering in the following set: {Random, BFT, DFT, RCM, MCA}, as introduced in Appendix G. Following [Jo et al., 2022], we select the best models based on the lowest validation FCD.

To implement the canonical orderings, we use the SciPy library [Virtanen et al., 2020] for the BFT, DFT, and RCM variants and the RDKit library [Landrum et al., 2006] for the MCA variant.

## F.4  BASELINES

We compare PGCs with various intractable graph DGMs. MoFlow [Zang and Wang, 2020] is a one-shot, normalizing flow model, a composition of (several layers of) two types of invertible, affine transformations. The first one models nodes

Table 3: *Hyper-parameters of the $\pi$PGCs variants*. The meaning of the hyper-parameters is as follows: $n_l$ is the number of layers, $n_S$ is the number of sum units, $n_I$ is the number of input units, $n_R$ is the number of repetitions [Peharz et al., 2020b], and $n_c$ is the number of weights in the root sum unit, which controls the ability to capture the correlations between the nodes and edges (Figure 1). N and E stands for the node-PC and edge-PC, respectively.

| Dataset | $\pi$PGC | | $n_l$ | $n_S$ | $n_I$ | $n_R$ | $n_c$ |
|---|---|---|---|---|---|---|---|
| QM9 | BT | N | $\{1,2,3\}$ | $\{16,32,64\}$ | $\{16,32\}$ | - | $\{1,4,16,64,256,512\}$ |
| | | E | $\{3,4,5\}$ | $\{16,32,64\}$ | $\{16,32\}$ | - | |
| | LT | N | $\{1,2,3\}$ | $\{16,32,64\}$ | $\{16,32\}$ | - | $\{1,4,16,64,256,512\}$ |
| | | E | $\{3,4,5\}$ | $\{16,32,64\}$ | $\{16,32\}$ | - | |
| | RT | N | $\{1,2,3\}$ | $\{16,32,64\}$ | $\{16,32\}$ | $\{16,32,64\}$ | $\{1,4,16,64,256,512\}$ |
| | | E | $\{3,4,5\}$ | $\{16,32,64\}$ | $\{16,32\}$ | $\{16,32,64\}$ | |
| | RT-S | N | $\{1,2,3\}$ | $\{16,32,64\}$ | $\{16,32\}$ | $\{16,32,64\}$ | $\{1,4,16,64,256,512\}$ |
| | | E | $\{3,4,5\}$ | $\{16,32,64\}$ | $\{16,32\}$ | $\{16,32,64\}$ | |
| | HCLT | N | - | $\{64,128,256,512\}$ | - | - | $\{1,4,16,64,256,512\}$ |
| | | E | - | $\{64,128,256,512\}$ | - | - | |
| Zinc250k | BT | N | $\{2,3,4\}$ | $\{16,32,64\}$ | $\{16,32\}$ | - | $\{1,4,16,64,256,512\}$ |
| | | E | $\{2,4,6\}$ | $\{16,32,64\}$ | $\{16,32\}$ | - | |
| | LT | N | $\{2,3,4\}$ | $\{16,32,64\}$ | $\{16,32\}$ | - | $\{1,4,16,64,256,512\}$ |
| | | E | $\{2,4,6\}$ | $\{16,32,64\}$ | $\{16,32\}$ | - | |
| | RT | N | $\{2,3,4\}$ | $\{16,32,64\}$ | $\{16,32\}$ | $\{16,32,64\}$ | $\{1,4,16,64,256,512\}$ |
| | | E | $\{2,4,6\}$ | $\{16,32,64\}$ | $\{16,32\}$ | $\{16,32,64\}$ | |
| | RT-S | N | $\{2,3,4\}$ | $\{16,32,64\}$ | $\{16,32\}$ | $\{16,32,64\}$ | $\{1,4,16,64,256,512\}$ |
| | | E | $\{2,4,6\}$ | $\{16,32,64\}$ | $\{16,32\}$ | $\{16,32,64\}$ | |
| | HCLT | N | - | $\{128,256,512,1024\}$ | - | - | $\{1,4,16,64,256,512\}$ |
| | | E | - | $\{128,256,512,1024\}$ | - | - | |

conditionally on observed edges and uses relational graph convolutional network (RGCN) [Schlichtkrull et al., 2018] to parameterize the affine transformation. The second models edges and relies on a variant of the Glow model [Kingma and Dhariwal, 2018] to parameterize the affine transformation. Continuous samples from the latent space are then mapped to discrete samples in the observation (graph) space, applying the dequantization (and quantization) to convert between discrete and continuous graphs (and vice versa). GraphAF [Shi et al., 2020] is an autoregressive, normalizing flow model that also uses an affine transformation parameterized by the RGCN to map continuous latent samples to discrete graphs, again using dequantization to perform the conversion from discrete to continuous samples. GraphDF [Luo et al., 2021] is an autoregressive, normalizing flow that eliminates the negative effect of dequantization and uses discrete modulo shift transformations to directly perform the mapping between the discrete latent space and discrete observation (graph) space.

EDP-GNN [Niu et al., 2020], GDSS [Jo et al., 2022], and DiGress [Vignac et al., 2023] are one-shot, diffusion models that noise and denoise input data through the forward and backward diffusion processes, receptively. The forward process of EDP-GNN perturbs input data with a sequence of increasing noise perturbations, jointly training a noise-conditioned neural network to estimate the score function—the gradient of the log distribution with respect to its input graph—by minimizing the score matching objective. The backward process utilizes annealed Langevin dynamics, recursively updating the score function with decreasing noise perturbations. GDSS realizes the forward (and backward) process through a system of positive (and negative) time-step stochastic differential equations, using the continuous-time version of the score-matching objective to train the node and edge score networks. EDP-GNN and GDSS rely on the dequantiazation, thus operating in the continuous space. DiGress is a discrete denoising diffusion model that works directly with the discrete node and edge attributes. The forward process relies on a Markov transition kernel to successively noise the node and edge attributes with discrete edits. In contrast, the backward process trains a graph transformer network to predict a clean graph from its noisy version, minimizing the cross entropy between the true and predicted graph. The simplicity and $\mathbb{S}_n$-equivariance of the denoising networks in these models is utilized to ensure that the targeted distribution over graphs is $\mathbb{S}_n$-invariant. GraphARM [Kong et al., 2023] is a node-absorbing, autoregressive diffusion model that also operates directly in the discrete space. The forward process absorbs one node at each step, masking the node and the associated edges. This masking mechanism is repeated until all the nodes are absorbed and the graph becomes empty. The backward process recovers the input graph by jointly training a denoising network (which parametrizes the backward graph transition kernel) and a dedicated diffusion ordering network (which parameterizes a probability distribution over the graph ordering).

GraphEBM [Liu et al., 2021] is a one-shot, energy-based model which parametrizes the energy function with the RGCN, also relying on its $\mathbb{S}_n$-equivariance to make the resulting probability distribution over graphs $\mathbb{S}_n$-invariant. SPECTRE [Martinkus

et al., 2022] is a generative adversarial network that uses spectral decomposition to model a probability distribution over a graph conditionally on top-$k$ eigenvalues and eigenvectors.

## F.5 COMPUTATIONAL RESOURCES

The experiments were conducted on a computational cluster with 56 Tesla A100 40GB GPUs. The jobs performing the gridsearch over the hyper-parameters in Table 3 were scheduled by SLURM 23.02.2. We limited each job to a single GPU. The computational time was restricted to 4 hours. All jobs were finished within this limit.

## G CANONICAL GRAPH ORDERINGS

Figure 6 illustrates a normalized empirical distribution over adjacency matrices of unordered and four canonically ordered graphs. For example, the Random case is nearly uniform, lacking any informative structure that could allow us to sample meaningful graphs. On the other hand, the BFT, DFT, RCM, and MCA orderings capture substantially more structural information, leading to a higher chance of generating more realistic and valid graph samples. The reason is that, for some orderings (e.g., BFT), the entries close to the diagonal are more concentrated, whereas those far from the diagonal are nearly zero. These concentrated patterns are also easier to learn than the Random case.

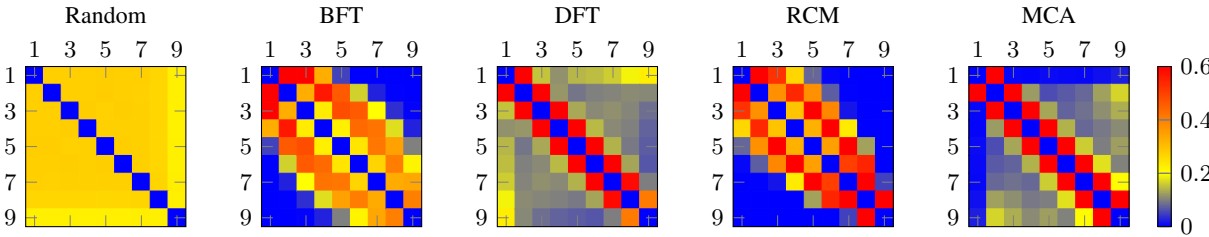

Figure 6: *The impact of different orderings on unnormalized empirical distribution over adjacency matrices.* The unnormalized empirical distribution is computed as the average of $\neg \mathbf{A}_{::1}$ for all graphs in the QM9 dataset [Ramakrishnan et al., 2014], where the entries of the matrix $\mathbf{A}_{::1}$ are equal to one if there is no edge between two nodes, and $\neg$ is the logical not. From left to right: all graphs in the dataset are randomly permuted (Random), ordered by the bread-first traversal (BFT), ordered by the deapth-first traversal (DFT), ordered by the reverse Cuthill-McKee (RCM) algorithm [Cuthill and McKee, 1969], ordered by the molecular canonicalization algorithm (MCA) which uses the domain knowledge to sort the graphs [Schneider et al., 2015].

## H ADDITIONAL RESULTS

**Unconditional molecule generation.** Figures 11-15 and Figures 16-20 provide examples of generated molecular graphs for the QM9 and Zinc250k datasets, respectively. The graphs show each variant of the $\pi$PGCs and each ordering from Appendix G. All the $\pi$PGCs variants deliver very similar performance across all orderings for the QM9 dataset. However, for the Zinc250k dataset, we observe that the $\pi$PGCs with the MCA and DFT orderings tend to generate rather elongated molecules that often contain no ring substructures. The BFT and RCM orderings, on the other hand, lead to more diverse and structured molecules with many ring substructures. The random ordering is more scattered and unusual compared to the other orderings.

**Conditional molecule generation.** Tables 4 and 5 display the molecular metrics for the conditional generation of molecules. In this experiment, we choose a specific molecular substructure to conditionally generate 10000 samples from the best models found in our original gridsearch (Appendix F.3). To compute the molecular metrics, we take the original splits of the QM9 and Zinc250k datasets that were used to train the models, and for each of these splits, we select only the molecules that contain the particular substructure.

Table 4: *Conditional generation on the QM9 datasets*. The mean value of the molecular metrics for various implementations of the $\pi$PGCs and different molecular subgraphs used to condition the model. The results are computed over five runs with different initial conditions. The 1st, 2nd, and 3rd best results are highlighted in colors. nAt and nBo are the average number of atoms and bonds, respectively, added to the newly generated part of the molecule. The parentheses in the Molecular scaffold column indicate the number of molecules in the training dataset containing this particular scaffold.

| Molecular scaffold | Model | Valid↑ | NSPDK↓ | FCD↓ | Unique↑ | Novel↑ | nAt | nBo |
|---|---|---|---|---|---|---|---|---|
| C1OCC=C1 (1295) | BT | 81.05 | 0.125 | 11.11 | 9.99 | 99.95 | 3.80 | 6.31 |
| | LT | 66.97 | 0.108 | 12.18 | 18.07 | 99.98 | 3.80 | 6.76 |
| | RT | 89.56 | 0.181 | 12.84 | 3.17 | 100.00 | 3.80 | 6.72 |
| | RT-S | 90.82 | 0.376 | 14.17 | 2.07 | 99.84 | 3.81 | 7.10 |
| | HCLT | 91.34 | 0.424 | 13.27 | 0.28 | 100.00 | 1.43 | 1.47 |
| N1NO1 (0) | BT | 74.27 | - | - | 64.17 | 100.00 | 5.80 | 6.73 |
| | LT | 45.28 | - | - | 83.45 | 100.00 | 5.80 | 7.63 |
| | RT | 68.12 | - | - | 46.91 | 100.00 | 5.80 | 7.05 |
| | RT-S | 82.13 | - | - | 39.42 | 100.00 | 5.80 | 6.96 |
| | HCLT | 52.81 | - | - | 2.20 | 100.00 | 2.74 | 2.25 |
| CCCO (59088) | BT | 87.96 | 0.054 | 5.17 | 38.40 | 97.33 | 4.80 | 5.93 |
| | LT | 73.10 | 0.064 | 6.24 | 48.26 | 97.99 | 4.79 | 5.86 |
| | RT | 88.42 | 0.048 | 4.75 | 26.90 | 95.86 | 4.80 | 5.96 |
| | RT-S | 94.42 | 0.052 | 4.68 | 19.29 | 94.48 | 4.80 | 5.88 |
| | HCLT | 75.04 | 0.324 | 16.59 | 1.16 | 98.05 | 1.89 | 1.71 |
| C1CNC1 (11421) | BT | 85.09 | 0.032 | 3.95 | 37.59 | 99.99 | 4.80 | 7.37 |
| | LT | 71.17 | 0.022 | 4.82 | 61.81 | 100.00 | 4.80 | 7.38 |
| | RT | 92.63 | 0.034 | 3.65 | 23.39 | 99.99 | 4.80 | 7.15 |
| | RT-S | 93.95 | 0.049 | 3.99 | 17.08 | 99.98 | 4.80 | 7.23 |
| | HCLT | 75.50 | 0.299 | 15.08 | 1.18 | 100.00 | 1.87 | 1.69 |
| CC(C)=O (11741) | BT | 69.58 | 0.118 | 11.21 | 23.29 | 99.92 | 4.80 | 6.04 |
| | LT | 54.26 | 0.127 | 10.88 | 31.96 | 99.93 | 4.80 | 6.28 |
| | RT | 76.77 | 0.145 | 12.55 | 13.29 | 99.92 | 4.80 | 6.13 |
| | RT-S | 80.83 | 0.112 | 11.32 | 10.17 | 99.94 | 4.80 | 6.27 |
| | HCLT | 55.66 | 0.393 | 16.12 | 1.06 | 95.97 | 1.85 | 1.75 |

Table 5: *Conditional generation on the Zinc250k dataset*. The mean value of the molecular metrics for various implementations of the $\pi$PGCs and different molecular subgraphs that are used to condition the model. The results are computed over five runs with different initial conditions. The 1st, 2nd, and 3rd best results are highlighted in colors. nAt and nBo are the average number of atoms and bonds, respectively, added to the newly generated part of the molecule. The parentheses in the Molecular scaffold column indicate the number of molecules in the training dataset containing this particular scaffold.

| Molecular scaffold | Model | Valid↑ | NSPDK↓ | FCD↓ | Unique↑ | Novel↑ | nAt | nBo |
|---|---|---|---|---|---|---|---|---|
| NS(=O)C1=CC=CC=C1 (0) | BT | 24.45 | - | - | 98.83 | 100.00 | 14.17 | 16.22 |
| | LT | 5.87 | - | - | 98.12 | 100.00 | 13.75 | 16.70 |
| | RT | 16.18 | - | - | 93.53 | 100.00 | 13.85 | 16.24 |
| | RT-S | 28.28 | - | - | 93.45 | 100.00 | 14.15 | 16.10 |
| | HCLT | 76.23 | - | - | 3.43 | 100.00 | 1.22 | 1.23 |
| CNC(C)=O (57329) | BT | 17.19 | 0.101 | 32.52 | 99.91 | 100.00 | 18.16 | 20.60 |
| | LT | 3.28 | 0.106 | 37.15 | 100.00 | 100.00 | 17.70 | 21.30 |
| | RT | 6.53 | 0.106 | 34.54 | 100.00 | 100.00 | 17.78 | 20.57 |
| | RT-S | 9.20 | 0.103 | 33.44 | 99.83 | 100.00 | 18.14 | 20.55 |
| | HCLT | 24.50 | 0.365 | 44.17 | 21.97 | 100.00 | 3.45 | 2.69 |
| O=C1CCCN1 (5287) | BT | 20.66 | 0.093 | 31.76 | 99.68 | 100.00 | 17.16 | 19.59 |
| | LT | 4.88 | 0.058 | 35.57 | 99.95 | 100.00 | 16.72 | 20.75 |
| | RT | 7.94 | 0.062 | 32.59 | 99.43 | 100.00 | 16.84 | 19.86 |
| | RT-S | 18.57 | 0.076 | 29.63 | 99.72 | 100.00 | 17.14 | 19.62 |
| | HCLT | 75.59 | 0.431 | 45.75 | 9.90 | 100.00 | 3.50 | 3.50 |
| C1CCNCC1 (28223) | BT | 27.52 | 0.066 | 28.70 | 99.91 | 100.00 | 17.16 | 19.67 |
| | LT | 7.45 | 0.064 | 32.42 | 100.00 | 100.00 | 16.71 | 19.66 |
| | RT | 13.41 | 0.054 | 31.01 | 99.79 | 100.00 | 16.80 | 19.82 |
| | RT-S | 25.47 | 0.060 | 27.89 | 99.88 | 100.00 | 17.14 | 20.28 |
| | HCLT | 44.30 | 0.335 | 40.61 | 16.08 | 99.95 | 3.50 | 3.06 |
| NS(=O)=O (15344) | BT | 24.72 | 0.129 | 38.56 | 99.88 | 100.00 | 19.16 | 21.45 |
| | LT | 3.16 | 0.120 | 47.55 | 100.00 | 100.00 | 18.69 | 22.45 |
| | RT | 12.90 | 0.121 | 41.90 | 99.93 | 100.00 | 18.83 | 21.41 |
| | RT-S | 31.29 | 0.145 | 36.74 | 98.60 | 100.00 | 19.15 | 21.26 |
| | HCLT | 0.51 | 0.595 | 58.08 | 11.76 | 100.00 | 3.66 | 2.09 |

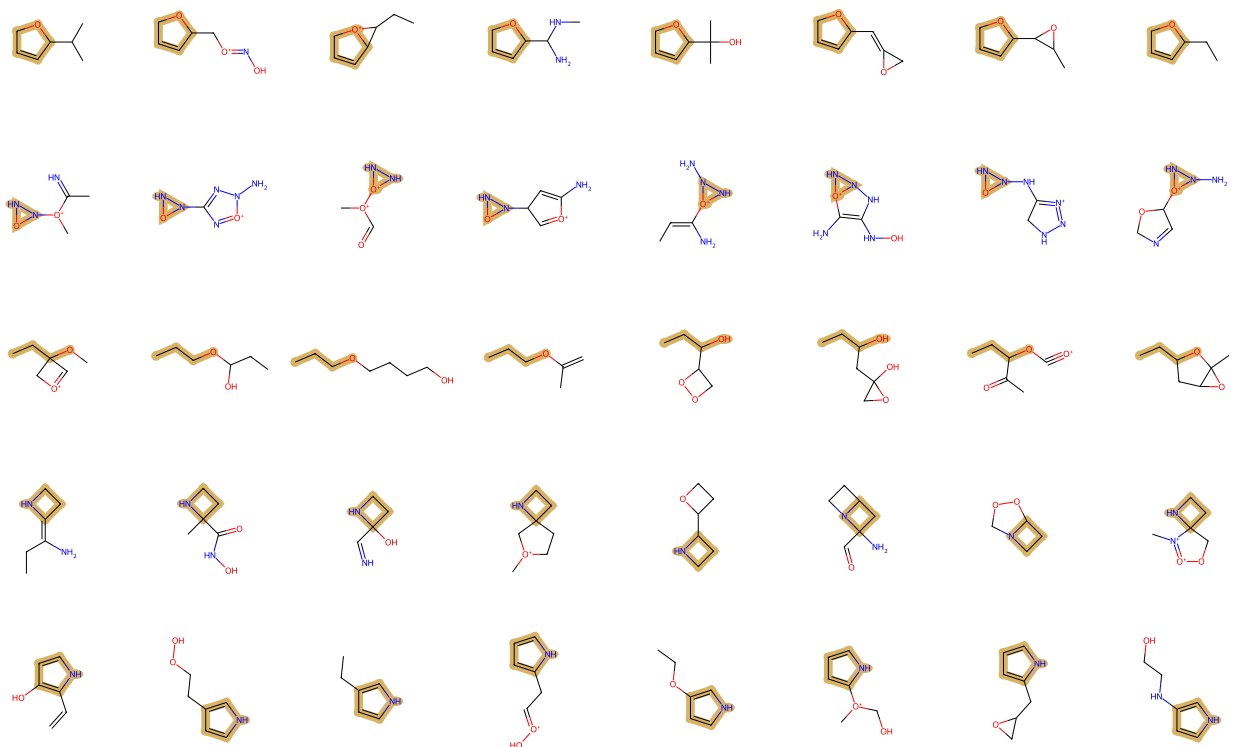

Figure 7: *Conditional generation on the QM9 dataset.* The yellow area highlights the known part of the molecule. There is one such known part per row. Each column corresponds to a new molecule generated conditionally on the known part. The samples were produced by the RT-S variant of $\pi$PGCs relying on the BFT ordering.

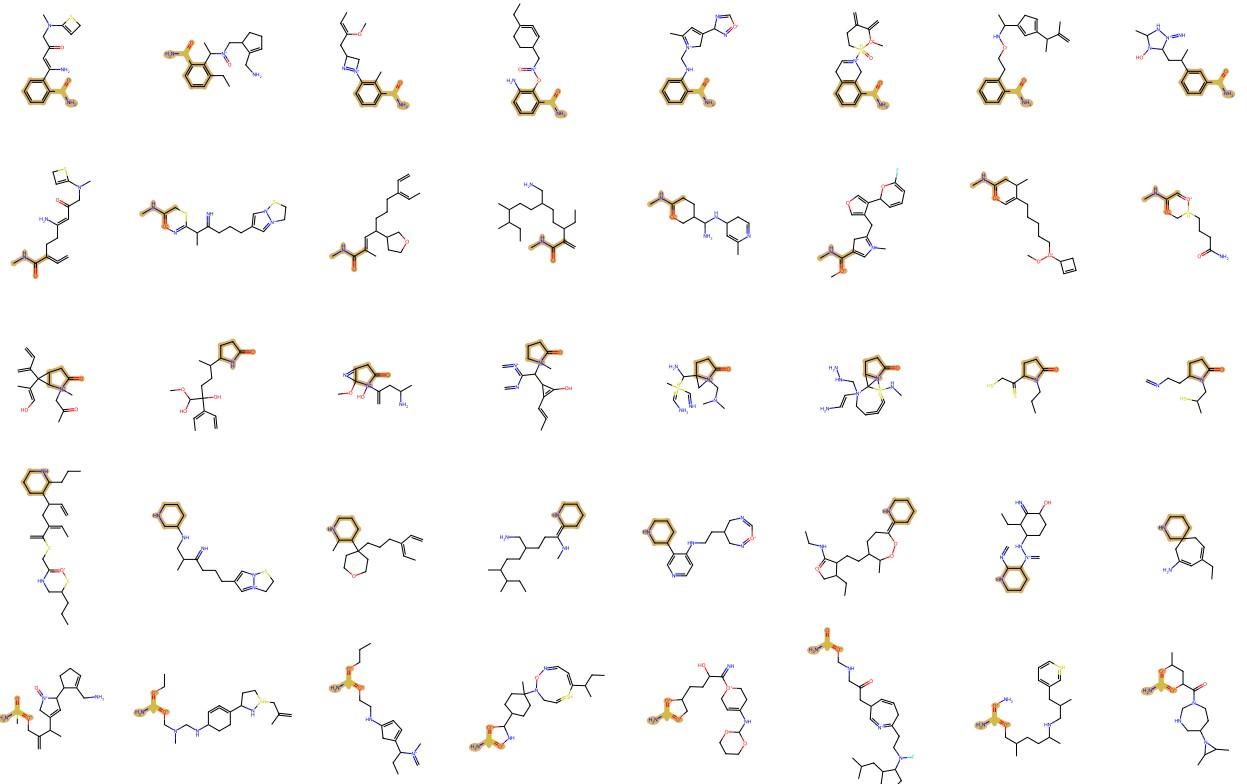

Figure 8: *Conditional generation on the Zinc250k dataset.* The yellow area highlights the known part of the molecule. There is one such known part per row. Each column corresponds to a new molecule that is generated conditionally on the known part. The samples were produced by the RT-S variant of $\pi$PGCs relying on the BFT ordering.

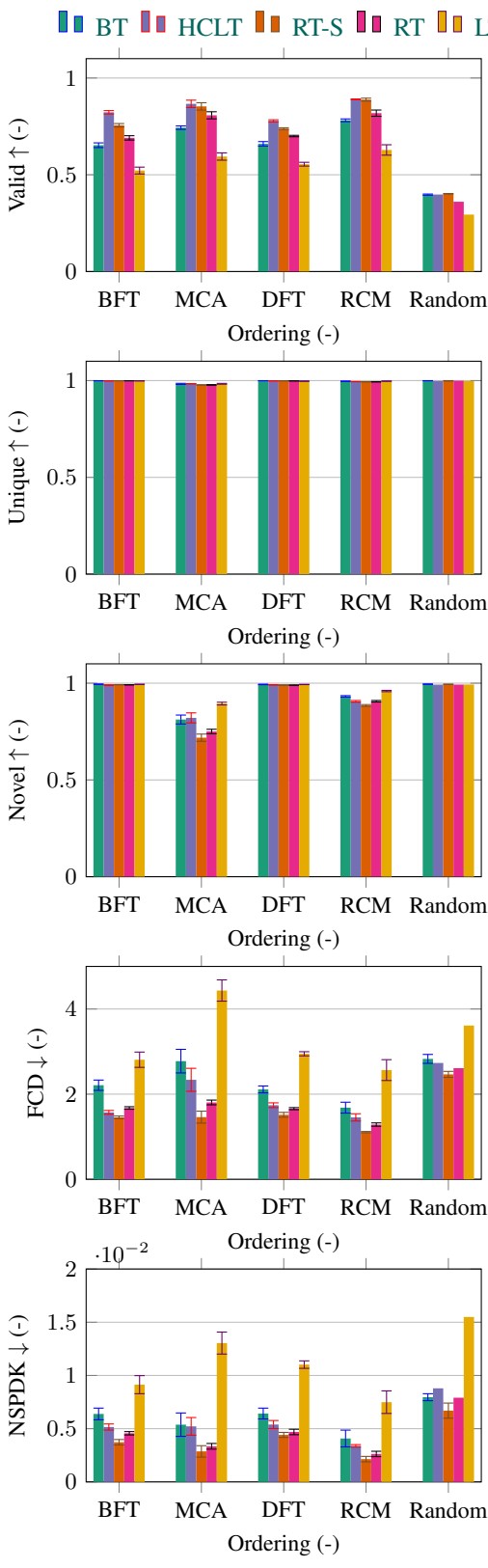

Figure 9: $\pi$*PGCs for different orderings on the QM9 dataset.* The mean (bar) and standard deviation (error bounds) of the molecular metrics for various implementations of the $\pi$PGCs that rely on the sorting to ensure the $\mathbb{S}_n$-invariance. The results are computed over five runs with different initial conditions.

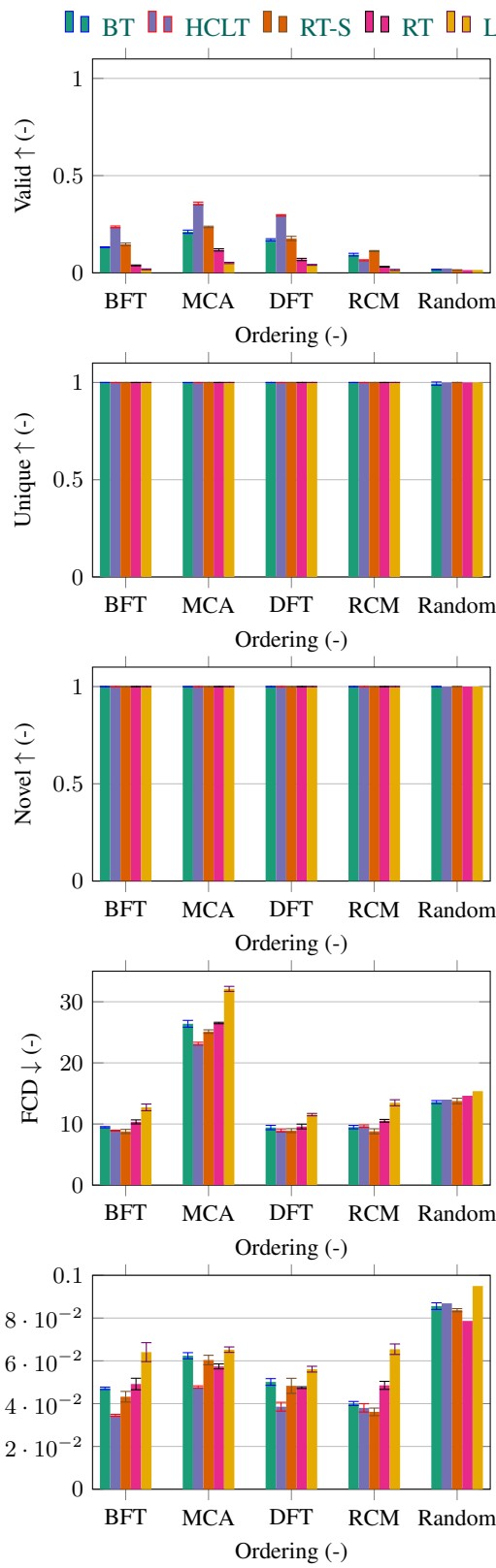

Figure 10: $\pi$*PGCs for different orderings on the Zinc250k dataset.* The mean (bar) and standard deviation (error bounds) of the molecular metrics for various implementations of the $\pi$PGCs that rely on the sorting to ensure the $\mathbb{S}_n$-invariance. The results are computed over five runs with different initial conditions.

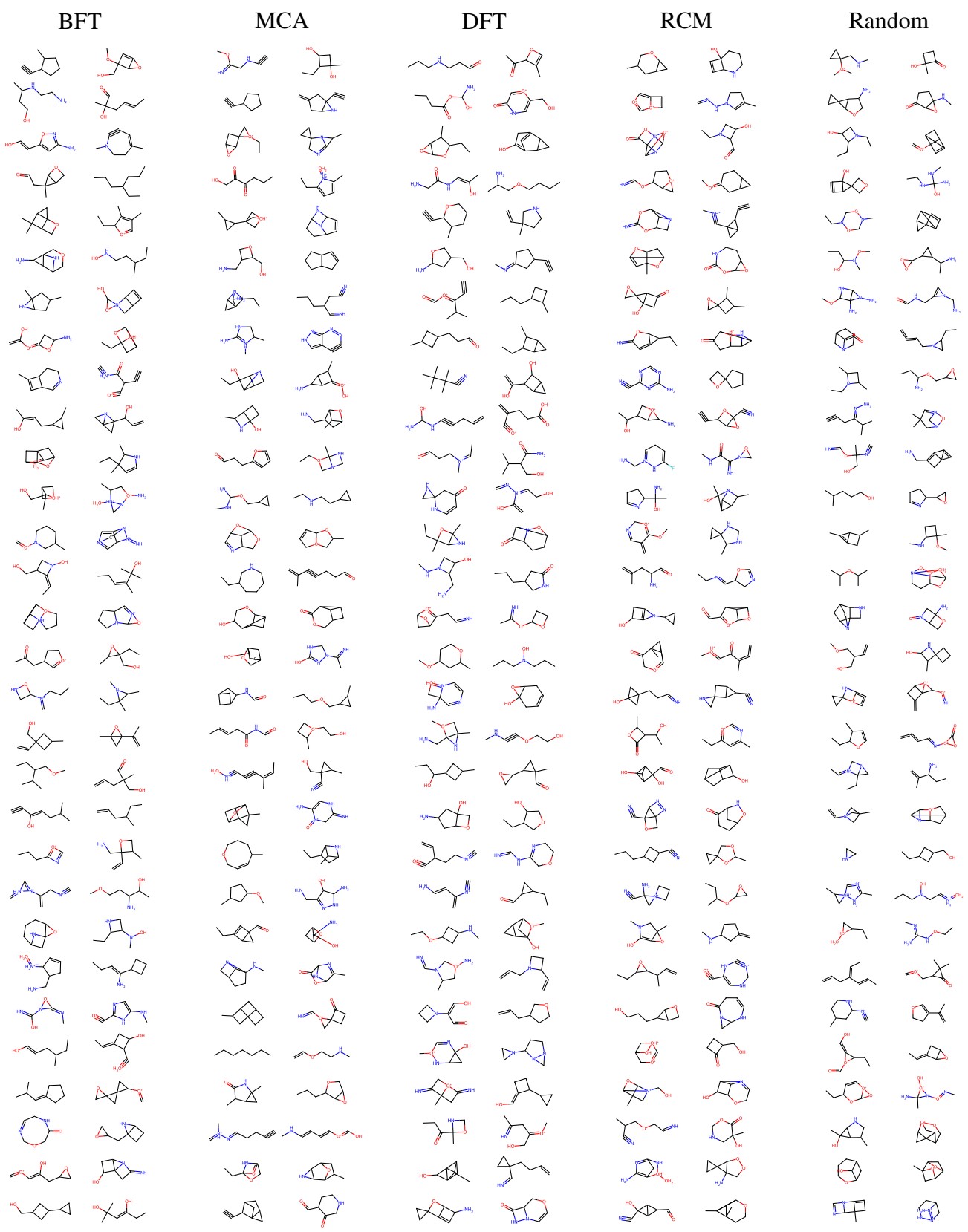

Figure 11: *Unconditional generation on the QM9 dataset.* Samples of molecular graphs for the BT variant of the $\pi$PGCs and different canonical orderings presented in Appendix G. Invalid molecules were rejected during the sampling.

Figure 12: *Unconditional generation on the QM9 dataset.* Samples of molecular graphs for the HCLT variant of the $\pi$PGCs and different canonical orderings presented in Appendix G. Invalid molecules were rejected during the sampling.

BFT          MCA          DFT          RCM          Random

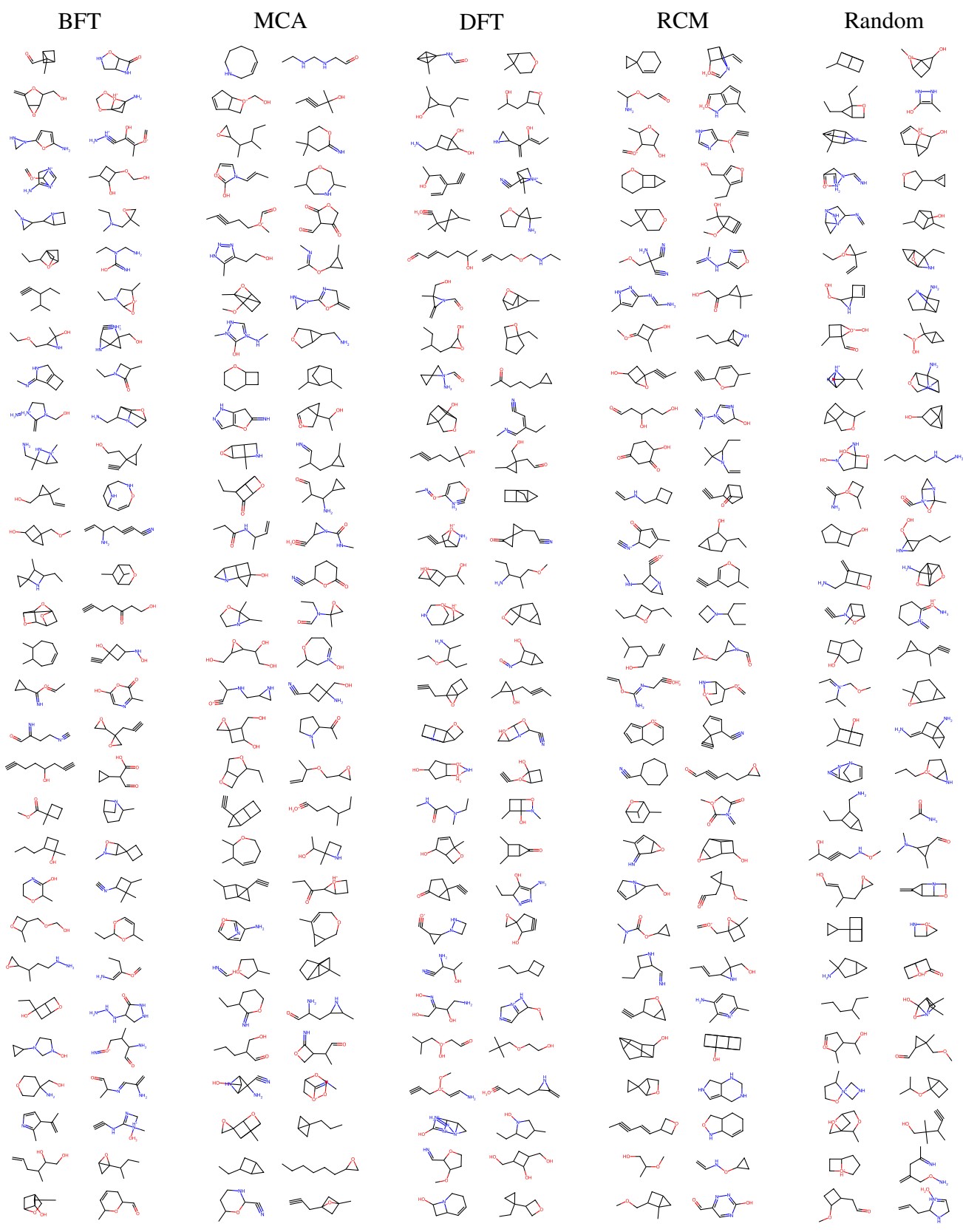

Figure 13: *Unconditional generation on the QM9 dataset.* Samples of molecular graphs for the RT-S variant of the $\pi$PGCs and different canonical orderings presented in Appendix G. Invalid molecules were rejected during the sampling.

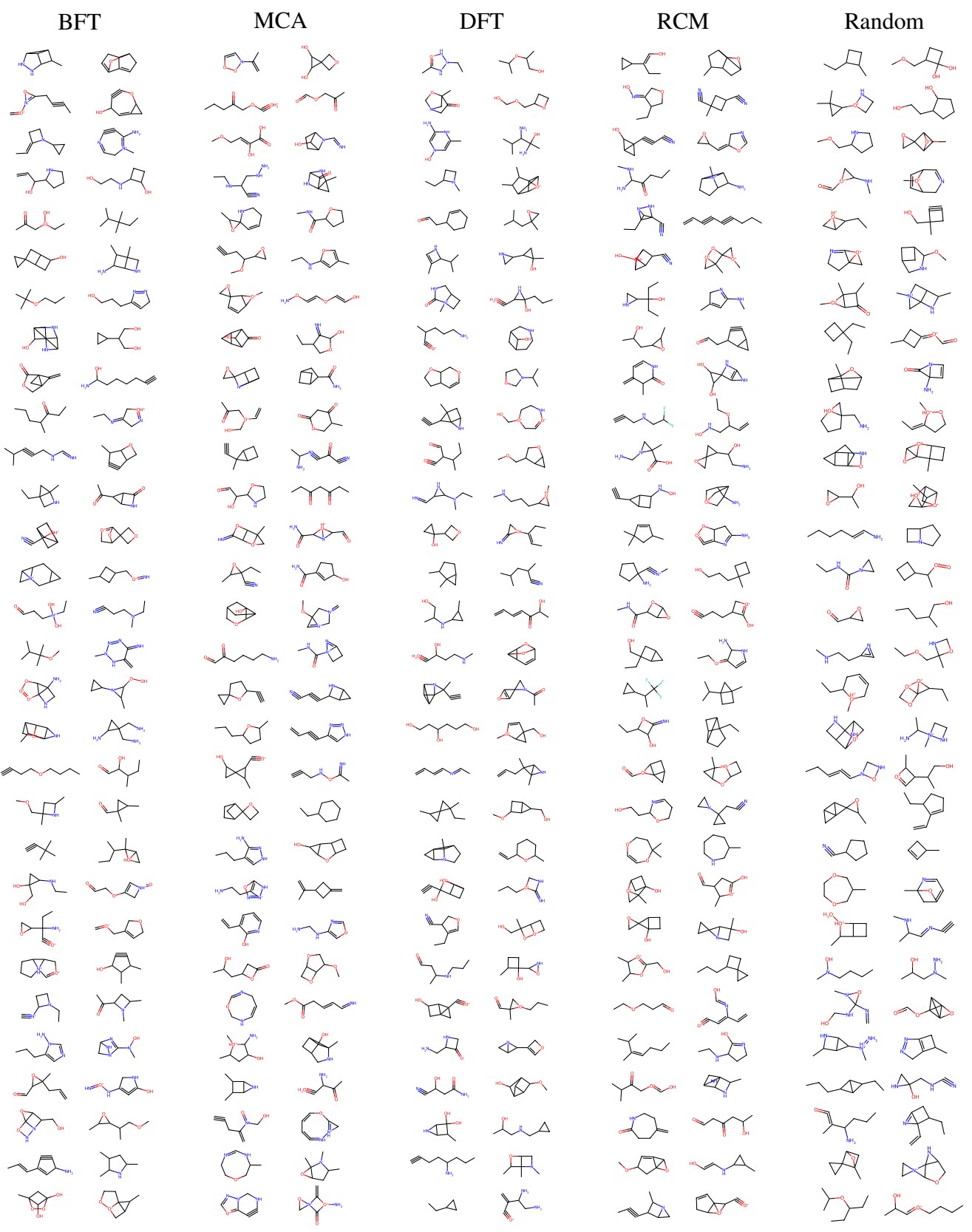

Figure 14: *Unconditional generation on the QM9 dataset.* Samples of molecular graphs for the RT variant of the $\pi$PGCs and different canonical orderings presented in Appendix G. Invalid molecules were rejected during the sampling.

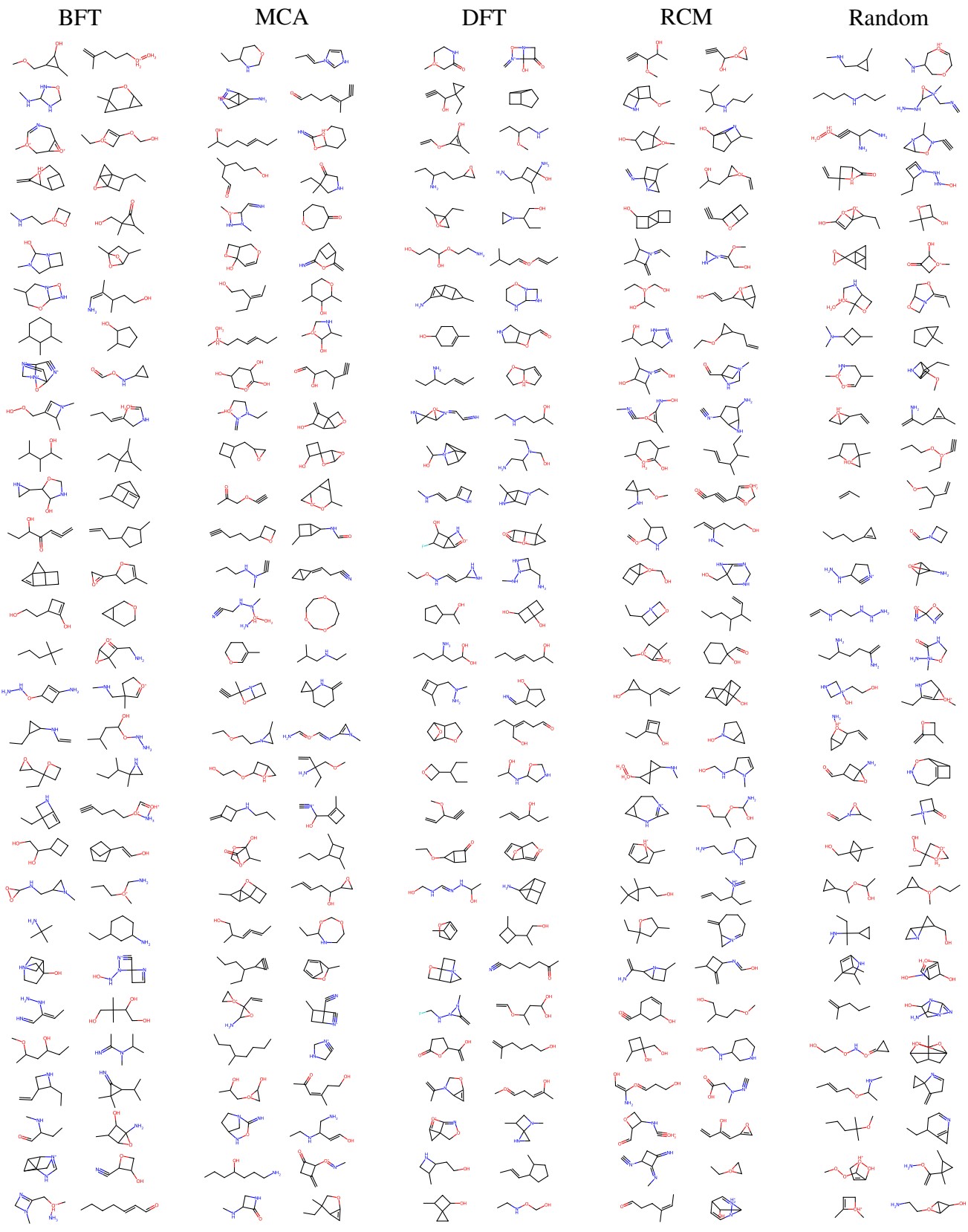

Figure 15: *Unconditional generation on the QM9 dataset.* Samples of molecular graphs for the LT variant of the $\pi$PGCs and different canonical orderings presented in Appendix G. Invalid molecules were rejected during the sampling.

Figure 16: *Unconditional generation on the Zinc250k dataset.* Samples of molecular graphs for the BT variant of the $\pi$PGCs and different canonical orderings presented in Appendix G. Invalid molecules were rejected during the sampling.

Figure 17: *Unconditional generation on the Zinc250k dataset.* Samples of molecular graphs for the HCLT variant of the $\pi$PGCs and different canonical orderings presented in Appendix G. Invalid molecules were rejected during the sampling.

Figure 18: *Unconditional generation on the Zinc250k dataset.* Samples of molecular graphs for the RT-S variant of the $\pi$PGCs and different canonical orderings presented in Appendix G. Invalid molecules were rejected during the sampling.

Figure 19: *Unconditional generation on the Zinc250k dataset.* Samples of molecular graphs for the RT variant of the $\pi$PGCs and different canonical orderings presented in Appendix G. Invalid molecules were rejected during the sampling.

Figure 20: *Unconditional generation on the Zinc250k dataset.* Samples of molecular graphs for the LT variant of the $\pi$PGCs and different canonical orderings presented in Appendix G. Invalid molecules were rejected during the sampling.