# OpenReview forum: "Probabilistic Graph Circuits: Deep Generative Models for Tractable Probabilistic Inference over Graphs"
_auai.org/UAI/2025/Workshop/TPM — TPM 2025_

### Official Review · Reviewer_VrPA · 2025-06-13
**A very well written paper about expressive probabilistic circuits on graph data**

**Rating:** 3

**Review:**

The paper introduces a family of probabilistic circuits (PCs) over graph data (e.g., molecules), that satisfy a certain number of desiderata. These includes being expressive w.r.t. other deep generative models on some benchmarks, as well as to model distributions of graphs with different sizes while being permutation invariant.

Overall, I have found the paper well written and easy to follow. In particular, I have really appreciated the introduction, which states very clearly the problems and the proposed contributions of the paper.

I believe there are a few issues, mainly regarding the notation and presentation, which I list below:
- Definition 3. $G_u$ is firstly defined as a sub-graph, i.e., a pair, but then it is assigned to the sets of nodes/edges. I found this confusing.
- It is not immediately clear from the definition of the $S_n$-invariant PGCs, Proposition 3 (conditional i.i.d. assumptions), and the paragraphs below to understand what is their structure in practice. However, the nice Figure 5 in the appendix help for this issue.

As a minor suggestion, I would have preferred seeing an example of canonical ordering in the main text, as to understand why its choice is particularly important in these models.

Regarding the empirical section, the authors carry out experiments regarding anomaly detection of molecules. To my understanding, they train a model over molecules with 5 atoms, and assume that any molecule with 6 atoms is an anomaly. The anomaly detection results are then shown in Figure 2 (right), which I think highlights extremely good performances for anomaly detection. However, I believe this setting can be made much more interesting and challenging by instead considering _impossible molecules_ as anomalous, e.g., impossible for physical/chemical constraints.

In conclusion, I believe this is a very interesting paper for the TPM community and recommend acceptance.

---

### Official Review · Reviewer_6Wz3 · 2025-06-17
**Probabilistic Graph Circuits: Deep Generative Models for Tractable Probabilistic Inference over Graphs**

**Rating:** 2

**Review:**

This paper studies the problem of modeling probability distributions over graphs, and proposes probabilistic graph circuits (PGCs) to balance the tradeoff between tractable inference and expressing permutation-invariant distributions. The paper was recently accepted to UAI 2025, is overall well written, and relevant to TPM audience. However, the proposed approach seems to be closely related to GraphSPNs [Papež et al., 2024] presented at TPM 2024, but the contributions of this work in relation to prior work are not clearly discussed.